# COPF: An Online Framework for Deployment-Stable Counterfactual Fairness in Evolving Graphs

**Sheng'en Li** [1]   **Dongmian Zou** [1]

## Abstract

Online link recommendation on evolving graphs is *performative*: by choosing which candidate links to show users, the system changes which links form and what feedback it later observes. Consequently, fairness estimates from logged outcomes can be misleading and may drift after deployment when the recommendation policy is updated. We introduce **COPF** (**C**ounterfactual **O**nline **P**erformative **F**airness), a decision-layer framework for deployment-stable fairness monitoring and control in online link recommendation. COPF (i) defines group-level opportunity gaps over exposure (shown vs. not shown) counterfactuals, (ii) makes them estimable by explicit exploration and by logging the probability (propensity) that each candidate is shown, and (iii) audits and controls fairness using residual outcome indistinguishability (OI) over a configurable auditor family with graph-aware doubly robust (GA-DR) estimators. We provide a noisy transfer theorem showing that Residual-OI on estimated GA-DR residuals implies bounds on exposure-counterfactual group gaps under temporal mixing and bounded local interference, and we instantiate an online multicalibration auditor together with a primal–dual controller. Experiments on two TGB streams and a controlled synthetic bipartite stream show that COPF reduces worst-case spikes in exposure-counterfactual group disparities with modest impact on ranking utility. Our code is available at https://github.com/lsnnnnnnnn/COPF.

[1]Zu Chongzhi Center and DIRC, Duke Kunshan University, Kunshan, China. Correspondence to: Dongmian Zou <dongmian.zou@duke.edu>.

*Proceedings of the 43rd International Conference on Machine Learning*, Seoul, South Korea. PMLR 306, 2026. Copyright 2026 by the author(s).

## 1. Introduction

Link prediction on temporal graphs predicts which edge will appear next given an evolving graph. It is often studied as an offline forecasting task (Rossi et al., 2020; Huang et al., 2023; Béres et al., 2019). However, in recommendation-driven settings (Su et al., 2016; Santos et al., 2021), each round the platform surfaces a set of candidate node pairs (dyads) to a user (e.g., "who to follow"), and many edges can form only after the user has been *exposed* to the suggested dyad (Ferrara et al., 2022). This sequential exposure and decision process turns forecasting into an *online* decision problem, closely related to contextual bandits for recommendation and ranking (Ban et al., 2024).

A key difficulty is that the system's decisions affect the data it will observe next. Empirically, link recommendations can reshape network structure and attention, increasing triadic closure (friends-of-friends becoming friends) (Su et al., 2016) and amplifying popularity inequality (Ferrara et al., 2022). More broadly, learning and evaluating on logs produced by a recommender means the observed outcomes reflect both user preferences and which candidates the platform chose to show (Chaney et al., 2018). If a group is shown fewer (or different) opportunities, the platform will also observe fewer positive outcomes for that group, which can reinforce future updates. As a result, fairness metrics computed directly on the realized log can be misleading.

Crucially, the shift can be self-induced. When the platform updates its recommendation policy (e.g., the ranking model), it changes what users see, which links can form, and what training examples are collected, so the data distribution can shift and fairness can drift after deployment (D'Amour et al., 2020; Liu et al., 2018). This phenomenon, where deploying a predictor changes the data-generating process it is evaluated on, i.e., the system changes the data it later learns from, is called *performative prediction* (Perdomo et al., 2020; Hardt & Mendler-Dünner, 2025). We call a fairness measure *deployment-stable* if it remains meaningful and comparable under such deployment-driven shifts, so fairness scores computed under different policies can be compared. Practically, we define it in terms of the counterfactual effect (Coston et al., 2020) of showing a candidate link: for each candidate, compare what would happen if

it were shown versus not shown, and then compare these effects across groups.

We present **COPF**, an end-to-end framework for deployment-stable fairness in online link recommendation on evolving graphs. COPF combines: (i) an online prequential protocol with exploration and propensity logging (Swaminathan & Joachims, 2015; Wang et al., 2022), which makes counterfactual auditing identifiable; (ii) graph-aware, self-normalized doubly robust estimators that remain consistent under temporal dependence and local network interference; and (iii) a certificate-and-control layer that audits *residual outcome indistinguishability* over rich auditor families, including optional Any-Kernel style kernelization (Dwork et al., 2025), and uses lightweight online primal–dual updates to steer exposure. Our primary fairness target is *benefit parity*: parity of the treatment effect of exposure across protected groups. To avoid achieving parity by simply withholding exposure, we add a minimum-effect guardrail and report within-group counterfactual calibration as a complementary diagnostic. Overall, we aim to maximize ranking utility while controlling these deployment-stable counterfactual disparities online.

**Conflict of Interest Disclosure.** The authors declare no financial conflicts of interest related to this work.

## 2. Related Work

**Performative prediction** Empirical studies have long documented that recommendation policies reshape the data they later train on. Su et al. (2016) analyzed Twitter's "Who to Follow" and showed that friend-of-friend recommendations increase triadic closure and amplify popularity inequality. Chaney et al. (2018) showed that training and evaluating on data collected under recommendations can homogenize behavior without improving utility. Related feedback mechanisms have been proposed in other decision domains such as predictive policing (Ensign et al., 2018) and selection settings where static fairness constraints can have delayed impacts (Liu et al., 2018; D'Amour et al., 2020). Perdomo et al. (2020) later formalized this type of endogenous distribution shift as *performative prediction*, and subsequent works studied stable points and learning dynamics (Perdomo et al., 2020; Mendler-Dünner et al., 2020; Brown et al., 2022; Hardt & Mendler-Dünner, 2025). Importantly for our setting, Mishler & Dalmasso (2022) showed that many observable fairness measures can be satisfied at training time but become unfair after deployment in performative environments, motivating counterfactual definitions.

Recent work also tracks how recommendation fairness evolves over time in dynamic social networks and studies counterfactual evolutions under alternative network trajectories (Cao et al., 2025), highlighting that fairness can drift

even when the underlying model class is fixed. COPF is complementary: rather than post-hoc analysis of logged snapshots, we target *deployment-time* comparability by defining fairness on exposure counterfactuals and making these quantities identifiable from an online, propensity-logged stream.

**Online link prediction metrics and methods** Outcome indistinguishability (OI) (Dwork et al., 2021) and multicalibration (Hébert-Johnson et al., 2018) enforce small residual errors across many subpopulations. Dwork et al. (2025) develop online OI for evolving graphs via the *Any-Kernel* method, enabling efficient (potentially infinite) kernelized auditor families. COPF adopts this auditing perspective, but (i) audits *exposure-counterfactual* residuals identified from propensity-logged exploration rather than realized outcomes, and (ii) couples auditing with causal identification and online exposure control.

**Counterfactual fairness, off-policy evaluation, and interference in networks.** Counterfactual fairness (Kusner et al., 2017) and counterfactual evaluation of predictive metrics (Coston et al., 2020) reason about decisions via potential outcomes, but identifiability can fail from observational logs without additional structure (Wu et al., 2019). COPF enforces overlap through explicit exploration and propensity logging, enabling doubly robust off-policy estimation on the resulting stream (Dudík et al., 2011; Swaminathan & Joachims, 2015); this is complementary to model-based approaches that learn counterfactual outcome distributions (Ma et al., 2026). Because exposure in social graphs can induce spillovers, we work under temporal dependence and bounded local interference assumptions (Ferrara et al., 2022; Santos et al., 2021; Su et al., 2016) and provide transfer guarantees in this setting. Finally, unlike counterfactual paradigms that hypothetically intervene on protected attributes (Bynum et al., 2024), COPF intervenes on exposure decisions (a platform action).

**Fairness in graph learning and recommender systems.** A broad literature studies fairness for link prediction and recommendation using various bias mitigation strategies. Adversarial objectives and regularization terms are commonly employed to enforce independence between learned representations and sensitive attributes (Dai & Wang, 2020; Masrour et al., 2020). Alternatively, other approaches focus on reweighting or modifying the graph topology, such as via biased edge dropout, to disrupt the propagation of bias (Li et al., 2022; Islam et al., 2021; Spinelli et al., 2021). Graph-specific causal (Sánchez-Martin et al., 2022) and counterfactual formulations (Ma et al., 2022) have also been explored for fair graph representation learning (Guo et al., 2023; Khajehnejad et al., 2022; Deldjoo et al., 2024). Unlike these approaches, COPF focuses on deployment-time

stability and can be applied on top of such backbones.

## 3. Problem Setup

We study online link recommendation on an evolving graph. Let $G_{\leq t} = (V, E_{\leq t})$ denote the pre-decision graph at round $t$. At each $t = 1, \ldots, T$, the stream reveals a source $u_t \in V$ and a ground-truth destination $v_t^\star \in V$. We form a candidate destination set $C_t = \{v_t^\star\} \cup N_t$, where $N_t$ contains up to $N$ negatives sampled online from a fixed pool (excluding $v_t^\star$). All features and structural summaries are computed only from $G_{\leq t}$.

For each candidate $v \in C_t$, $W_t(u_t, v)$ denotes the pre-exposure decision context for the dyad: it contains the dyad features $(\phi_t(u_t, v), G_t^{\mathrm{loc}}(u_t, v))$ and the slate-policy inputs used by $\pi_t$, such as $C_t$ and $\hat{p}_t(u_t, \cdot)$. A backbone predictor produces a score $\hat{p}_t(u_t, v) \in (0, 1)$ for every $v \in C_t$. An exposure policy selects a slate $D_t \subseteq C_t$ of size $K$; here we write $D_t(v) = \mathbf{1}\{v \in D_t\}$.

Because we later perform counterfactual evaluation, we log for each $v \in C_t$ its *marginal entry propensity*

$$e_t(v) := \Pr_{D_t \sim \pi_t(\cdot|H_t)} (v \in D_t),$$

where $H_t$ denotes the information available when constructing the slate (e.g., $H_t = (G_{\leq t}, u_t, C_t, \hat{p}_t(u_t, \cdot))$). In our Top-$K$ stochastic policy, $e_t(v)$ is exact for uniform exploration and is estimated for the score-based Plackett–Luce component via Monte Carlo (128 samples by default); we log a clipped estimate $\hat{e}_t(v)$ for numerical stability. True overlap is policy-enforced rather than created by clipping: under the mixture policy with uniform exploration probability $\epsilon$, every candidate has marginal inclusion probability at least $\epsilon K / |C_t|$, and also has positive non-exposure probability when $K < |C_t|$. Approximation and clipping errors are absorbed into the propensity nuisance term $\varepsilon_e$ in Lemma 4.2.

We use potential outcomes $Y_t^{(1)}(u_t, v) \in \{0, 1\}$ and $Y_t^{(0)}(u_t, v) \in \{0, 1\}$ for whether the link $(u_t, v)$ would form if $v$ were exposed or not, respectively. The observed outcome satisfies the standard consistency relation

$$Y_t(u_t, v) = D_t(v) \, Y_t^{(1)}(u_t, v) + (1 - D_t(v)) \, Y_t^{(0)}(u_t, v).$$

With local spillovers, $Y_t^{(a)}(u_t, v)$ denotes the own-exposure, policy-marginal counterfactual: set $D_t(v) = a$ and average the remaining local exposures under the logged policy conditional on $W_t$. In our experiments we use a banditized feedback model where only exposed candidates yield feedback: $Y_t(u_t, v) = \mathbf{1}\{v = v_t^\star\} D_t(v)$.

**Protected attribute and slices.** We evaluate fairness with respect to a (possibly synthetic) protected attribute. Let $A_i \in \mathcal{A}$ denote the attribute of node $i$, taking values in a finite set $\mathcal{A}$. To attach a group label to each candidate dyad $(u_t, v)$, we fix an attribution rule and define

$$A_t(u_t, v) = \begin{cases} A_{u_t}, & \text{(source-side auditing)} \\ A_v, & \text{(destination-side auditing)}. \end{cases}$$

When conditioning in expectations we write $A$ for this dyad label. For auditing and control we index subpopulations (slices) by group membership together with score buckets and, optionally, a discrete structural role. Specifically, for each group $s$ we form equal-mass score buckets $(I_{s,b})_{b=1}^{B_{\mathrm{bucket}}}$ of the current scores $\hat{p}_t(u_t, v)$ (e.g., via group-conditional quantiles within each logging window), and let $R_t(u_t, v) \in \mathcal{R}$ summarize a role derived from $G_t^{\mathrm{loc}}$. A typical slice corresponds to the indicator

$$h_{s,b,\rho}(u_t, v) = \mathbf{1}\{A_t(u_t, v) = s, \hat{p}_t(u_t, v) \in I_{s,b}, R_t(u_t, v) = \rho\},$$

dropping the $R_t$ term when roles are disabled.

### 3.1. Prequential logging protocol and identifiability

To make decision-counterfactual quantities identifiable from a single stream, we adopt an online prequential protocol (OPP) with explicit exploration and propensity logging:

- **OPP-0 (No leakage):** process events in time order; all features/statistics at $t$ use $G_{\leq t}$ before update.
- **OPP-1 (Online candidates):** construct $C_t$ prospectively (true destination plus feasible negatives sampled at $t$).
- **OPP-2 (Explore & log):** use a stochastic exposure policy with explicit exploration to ensure overlap; log marginal propensities $\hat{e}_t(v)$ for every $v \in C_t$.
- **OPP-3 (Update online):** update the backbone and (when used) nuisance outcome models sequentially, with online cross-fitting.
- **OPP-4 (Prequential audit & control):** estimate utility and counterfactual fairness online from the logged stream and, when enabled, update the decision-layer controller.

**Assumption 3.1** (Local identifiability on evolving graphs). For each round $t$ and candidate $v \in C_t$, let $W_t = W_t(u_t, v)$. **(i) Overlap.** There exists $\underline{e} > 0$ such that $\underline{e} \leq e_t(v) \leq 1 - \underline{e}$. **(ii) Local ignorability.** Conditional on $W_t$ (measurable before choosing the slate), the exposure indicator $D_t(v)$ is independent of the potential outcomes $(Y_t^{(0)}(u_t, v), Y_t^{(1)}(u_t, v))$. **(iii) Bounded dependence.** The process exhibits bounded local interference and is temporally mixing (e.g., $\beta$-mixing with mixing time $\tau_{\mathrm{mix}}$).

**Definition 3.2** (Graph-aware self-normalized doubly robust (GA-DR) pseudo-outcomes). Let $\mu_a(W) = \mathbb{E}[Y^{(a)} \mid W]$ and let $\hat{\mu}_{a,t}(W)$ be online nuisance estimates. Define arm propensities $\hat{e}_t^{(1)}(v) = \hat{e}_t(v)$ and $\hat{e}_t^{(0)}(v) = 1 - \hat{e}_t(v)$. For $a \in \{0, 1\}$, the doubly robust pseudo-outcome for candidate

$(u_t, v)$ is

$$\tilde{\Gamma}_t^{(a)}(u_t, v) = \hat{\mu}_{a,t}(W_t)$$
$$+ \frac{\mathbf{1}\{D_t(v) = a\}}{\hat{e}_t^{(a)}(v)}\big(Y_t(u_t, v) - \hat{\mu}_{a,t}(W_t)\big).$$

Given nonnegative graph-aware weights $w_t(u_t, v)$ (time decay and local-structure weighting), we estimate expectations with self-normalization over a window $\mathcal{W}$: $\widehat{\mathbb{E}}_{\mathrm{GA},\mathcal{W}}[Z] = \frac{\sum_{(t,v)\in\mathcal{W}} w_t(u_t,v) Z_t(u_t,v)}{\sum_{(t,v)\in\mathcal{W}} w_t(u_t,v)}$. We form estimated residuals $\hat{r}^{(0)} = \tilde{\Gamma}^{(0)} - \hat{p}$ and $\hat{r}^{(\Delta)} = (\tilde{\Gamma}^{(1)} - \tilde{\Gamma}^{(0)}) - \tau(W_t)$.

**Definition 3.3** (Decision-counterfactual fairness gaps). For each group $s \in \mathcal{A}$, let $\tau_s = \mathbb{E}\big[Y^{(1)} - Y^{(0)} \mid A = s\big]$ denote the average treatment effect of exposure. We evaluate: (i) **benefit parity** via the treatment-effect gap $g_{\mathrm{gap}}^{\mathrm{TE}} = \max_{s,s'} |\tau_s - \tau_{s'}|$; (ii) a **minimum-effect** guardrail $g^{\mathrm{Min}} = \max_s [\tau_{\min} - \tau_s]_+$ (for a chosen $\tau_{\min}$); and (iii) **within-group counterfactual calibration** $g_{\max}^{\mathrm{Cal},(a)}$ as a diagnostic: For equal-mass score buckets $(I_{s,b})_{b=1}^{B_{\mathrm{bucket}}}$, define the slice gap

$$g_{s,b}^{\mathrm{Cal},(a)} = \left| \mathbb{E}\big[Y^{(a)} \mid A = s, \hat{p} \in I_{s,b}\big] - \mathbb{E}\big[\hat{p} \mid A = s, \hat{p} \in I_{s,b}\big] \right|,$$

$$g_{\max}^{\mathrm{Cal},(a)} = \max_{s,b} g_{s,b}^{\mathrm{Cal},(a)}.$$

We report $g_{\max}^{\mathrm{Cal}} := g_{\max}^{\mathrm{Cal},(0)}$ (GA-DR plug-ins on rolling windows). For $a = 0$, Lemma 4.1 yields the equivalent slice form $g_{s,I_{s,b}}^{\mathrm{Cal}}$ used in certificates.

**Goal.** Given a backbone scorer and an OPP stream with propensity logs, our goal is to (a) provide identifiable online estimates (and certificates) of the above counterfactual gaps, and (b) adjust exposures online to keep $g_{\mathrm{gap}}^{\mathrm{TE}}$ and $g^{\mathrm{Min}}$ small while maintaining high utility.

## 4. Methodology

COPF is a *decision-layer* framework: it wraps a fixed (or continuously trained) backbone scorer $\hat{p}_t(u_t, v)$ with an online layer that (i) enforces identifiable logging through the OPP protocol in Sec. 3.1, (ii) estimates exposure-counterfactual quantities via GA-DR residuals (Def. 3.2), (iii) audits Residual-OI to produce finite-window certificates (Sec. 4.1), and (iv) updates lightweight score offsets and dual variables to meet counterfactual fairness targets with limited utility loss (Sec. 4.3). We emphasize benefit parity (small $g_{\mathrm{gap}}^{\mathrm{TE}}$) together with a minimum-effect guardrail ($g^{\mathrm{Min}}$) to avoid fairness-through-rationing, and report counterfactual calibration gaps as a diagnostic.

### 4.1. Residual-OI Auditing and Certificates

Following OI (Dwork et al., 2025) and multicalibration (Hébert-Johnson et al., 2018), we view an *auditor* as a bounded test function $h$ that selects a subpopulation (a "slice") and checks whether the average residual on that slice is close to zero. Operationally, to *audit* a residual sequence means to compute $\sup_{h\in\mathcal{H}} |\widehat{\mathbb{E}}[h\,r]|$ over a chosen auditor family $\mathcal{H}$; if this quantity is small, then no auditor in $\mathcal{H}$ can refute the claim that the residual is well-calibrated across those slices.

In COPF we audit *decision-counterfactual residuals* rather than observed outcomes. Using GA-DR pseudo-outcomes (Def. 3.2), we form two residual sequences: (i) a no-exposure residual $\hat{r}^{(0)} = \tilde{\Gamma}^{(0)} - \hat{p}$ used for multicalibration and calibration diagnostics, and (ii) an effect residual $\hat{r}^{(\Delta)} = (\tilde{\Gamma}^{(1)} - \tilde{\Gamma}^{(0)}) - \tau(W_t)$ used to control benefit parity. For the residual-only treatment-effect certificate below, the centering target is chosen to be group-mean invariant over the audited groups, e.g., a scalar $\bar{\tau}$ held fixed within the audit window. Lemma 4.1 shows that calibration gaps, and treatment-effect gaps under this centering, can be written as (ratios of) correlations between slice indicators and these residuals. Therefore, enforcing small residual correlations for all auditors in $\mathcal{H}$ yields certificates for the downstream counterfactual fairness gaps.

**Residual-OI (windowed, GA-weighted).** Let $\widehat{\mathbb{E}}_{\mathrm{GA},\mathcal{W}}[\cdot]$ denote the graph-aware (GA), self-normalized empirical average over the audit window $\mathcal{W}$ of length $L_{\mathrm{win}}$ (the same window used by our runner for both metrics and auditing), optionally using GA weights $w_t$. Given a residual sequence $(r_t)$ and an auditor class $\mathcal{H}$, define the (empirical) residual-OI gap

$$\widehat{\mathrm{OI}}_{\mathcal{W}}(r; \mathcal{H}) \triangleq \sup_{h\in\mathcal{H}} \left| \widehat{\mathbb{E}}_{\mathrm{GA},\mathcal{W}}\big[h_t\,r_t\big] \right|.$$

We say $\varepsilon$-Residual-OI holds on the window if $\widehat{\mathrm{OI}}_{\mathcal{W}}(r; \mathcal{H}) \leq \varepsilon$. In our implementation, $\mathcal{H}$ is instantiated as indicators of slices (group $\times$ score-bucket, and optionally structural roles), so $h_t \in \{0, 1\}$.

**Finite-sample certificates used in the runner.** Our fairness metrics are ratio-type quantities: a slice-specific numerator (a GA-weighted residual moment) normalized by the slice mass. Concretely, for calibration slices indexed by $(s, I)$,

$$g_{s,I}^{\mathrm{Cal}} = \frac{\left| \widehat{\mathbb{E}}_{\mathrm{GA},\mathcal{W}}[\mathbf{1}\{A = s, \hat{p} \in I\} r^{(0)}] \right|}{\widehat{\mathbb{E}}_{\mathrm{GA},\mathcal{W}}[\mathbf{1}\{A = s, \hat{p} \in I\}]}.$$

Thus, if we can upper bound the *numerator* uniformly over audited slices by $\varepsilon_0 + \beta_0$ (empirical worst-case gap plus a concentration radius), then dividing by the smallest audited slice mass $p_{\min}^{\mathrm{gb}}$ yields a uniform upper bound on the calibration error:

$$g_{\max}^{\mathrm{Cal}} \leq \frac{\varepsilon_0 + \beta_0}{p_{\min}^{\mathrm{gb}}}.$$

For treatment-effect parity under the same group-mean-invariant centering, the gap compares two groups, so the worst-case group difference is bounded by the sum of their (absolute) residual moments; this contributes a factor of 2:

$$g_{\text{gap}}^{\text{TE}} \leq \frac{2(\varepsilon_\Delta + \beta_\Delta)}{p_{\min}^{\text{g}}}.$$

Here $(\varepsilon_0, \beta_0)$ audits the residual $r^{(0)}$ over group×bucket slices, while $(\varepsilon_\Delta, \beta_\Delta)$ audits the residual $r^{(\Delta)}$ over group slices.

## 4.2. Theory: from plug-in Residual-OI to counterfactual gap bounds

COPF audits *estimated* DR residuals online, but our fairness criteria are defined on the *true* counterfactual residuals. The key technical question is therefore: when does Residual-OI on the plug-in residuals imply guarantees on the true counterfactual gaps? We summarize the main steps here and defer details to the appendix.

**Lemma 4.1** (Fairness gaps linearize to residuals). *Under overlap, for any group $s$ and bucket $I$ with $\Pr(A = s, \hat{p} \in I) > 0$,*

$$g_{s,I}^{\text{Cal}} = \frac{\left|\mathbb{E}[\mathbf{1}\{A = s, \hat{p} \in I\} r^{(0)}]\right|}{\Pr(A = s, \hat{p} \in I)}.$$

*If, in addition, the effect-centering target used for treatment-effect auditing is group-mean invariant over the audited groups,*

$$\mathbb{E}[\tau(W_t) \mid A = s'] = m_\tau \qquad \text{for every audited group } s',$$

*then for any groups $s_1, s_2$ with $\Pr(A = s_i) > 0$,*

$$g_{s_1,s_2}^{\text{TE}} = \left| \frac{\mathbb{E}[\mathbf{1}\{A = s_1\} r^{(\Delta)}]}{\Pr(A = s_1)} - \frac{\mathbb{E}[\mathbf{1}\{A = s_2\} r^{(\Delta)}]}{\Pr(A = s_2)} \right|.$$

*Moreover, for any group $s$ with $\Pr(A = s) > 0$,*

$$g_s^{\text{Min}} = \left[ \tau_{\min} - \left\{ \frac{\mathbb{E}[\mathbf{1}\{A = s\} r^{(\Delta)}]}{\Pr(A = s)} + \mathbb{E}[\tau(W_t) | A = s] \right\} \right]_+.$$

**Lemma 4.2** (Self-normalized GA-DR under mixing). *Under Assumption 3.1 with online cross-fitting and clipping, for any arm $a \in \{0, 1\}$ and audit window $\mathcal{W}$, uniformly over $h \in \mathcal{H}$,*

$$\left| \widehat{\mathbb{E}}_{\text{GA},\mathcal{W}}\left[ h(W_t) \tilde{\Gamma}_t^{(a)} \right] - \mathbb{E}_{\text{GA},\mathcal{W}}\left[ h(W_t) Y_t^{(a)} \right] \right|$$
$$\leq C_1 \varepsilon_e \varepsilon_\mu + C_2(\varepsilon_e^2 + \varepsilon_\mu^2) + C_3 \sqrt{\frac{\tau_{\text{mix}} \log |\mathcal{H}|}{W_{\text{eff}}}}.$$

**Assumption 4.3** (Online plug-in Residual-OI over $\mathcal{H}$). On the current GA-weighted audit window, an online auditing/multicalibration routine over $\mathcal{H}$ attains

$$\varepsilon_T \triangleq \sup_{h \in \mathcal{H}} \left| \widehat{\mathbb{E}}_{\text{GA},\mathcal{W}}[h(W_t) \hat{r}_t] \right| \leq C W_{\text{eff}}^{-1/2} \text{polylog}(W_{\text{eff}}),$$

with per-auditor confidence radii

$$\beta_T = O\left( \sqrt{\frac{\log |\mathcal{H}|}{W_{\text{eff}}}} \right).$$

*Remark* 4.4 (Modular plug-in certificates). Theorem 4.5 is modular: given any plug-in Residual-OI certificate $(\varepsilon_T, \beta_T)$ on the estimated GA-DR residuals, it yields bounds on the corresponding *true* counterfactual gaps (up to estimation and dependence terms). Assumption 4.3 is thus an input condition, not a proved rate for our budgeted active-auditor heuristic; analyzing such active-set dynamics under dependence is left to future work.

**Theorem 4.5** (Noisy residual-OI $\Rightarrow$ true residual control). *Let $\hat{r} \in \{\hat{r}^{(0)}, \hat{r}^{(\Delta)}\}$ be the GA-DR plug-in residual for a true counterfactual residual $r$. Suppose the plug-in estimation error is controlled as in Lemma 4.2, and suppose the windowed Residual-OI certificate satisfies*

$$\sup_{h \in \mathcal{H}} \left| \widehat{\mathbb{E}}_{\text{GA},\mathcal{W}}[h(W_t) \hat{r}_t] \right| \leq \varepsilon_T.$$

*Under Assumptions 3.1–4.3 and the windowed deviation bound in Theorem D.4, with high probability,*

$$\sup_{h \in \mathcal{H}} |\mathbb{E}_{\text{GA},\mathcal{W}}[h(W_t) r_t]| \leq \varepsilon_T + c_1 \varepsilon_e \varepsilon_\mu + c_2(\varepsilon_e^2 + \varepsilon_\mu^2)$$
$$+ c_3 \sqrt{\frac{\tau_{\text{mix}} \log |\mathcal{H}|}{W_{\text{eff}}}} + c_4 \beta_T.$$

**Corollary 4.6** (Residual-OI $\Rightarrow$ counterfactual fairness). *Under the group-mean-invariant effect centering in Lemma 4.1, for audited groups and group-bucket slices with masses at least $p_{\min}^{\text{g}}$ and $p_{\min}^{\text{gb}}$, respectively,*

$$g_{s_1,s_2}^{\text{TE}} \leq \frac{2(\varepsilon_T + E_T)}{p_{\min}^{\text{g}}}, \qquad g_{s,I}^{\text{Cal}} \leq \frac{\varepsilon_T + E_T}{p_{\min}^{\text{gb}}}.$$

*Moreover, writing $m_s = \mathbb{E}[\tau(W_t) \mid A = s]$,*

$$g_s^{\text{Min}} \leq \left[ \tau_{\min} - m_s + \frac{\varepsilon_T + E_T}{p_{\min}^{\text{g}}} \right]_+,$$

*where*

$$E_T = c_1 \varepsilon_e \varepsilon_\mu + c_2(\varepsilon_e^2 + \varepsilon_\mu^2) + c_3 \sqrt{\frac{\tau_{\text{mix}} \log |\mathcal{H}|}{W_{\text{eff}}}} + c_4 \beta_T.$$

In short, Theorem 4.5 shows that if we can make the plug-in GA-DR residuals indistinguishable to the audited family $\mathcal{H}$, then the *true* counterfactual gaps are small up to estimation and dependence terms; Corollary 4.6 then yields the mass-normalized certificates used by our runner (Sec. 4.1).

## 4.3. Objective and Online Constrained Optimization

COPF can be read as an online constrained optimization problem: maximize ranking utility while keeping the counterfactual gaps in Def. 3.3 below target tolerances. In implementation, we instantiate the constraints as a decision-layer controller. Concretely, we treat the backbone predictor $f_t$ as a black box that outputs a base probability $\hat{p}_t \in (0, 1)$ for each candidate edge. COPF then applies online logit adjustments before the exposure policy acts.

**Decision-layer parameterization.** For a candidate with dyad group label $s = A_t(u_t, v) \in \mathcal{A}$ and score-bucket index $b$, COPF forms the probability used for exposure as

$$\tilde{p}_t(v) = \sigma\big(\text{logit}(\hat{p}_t(u_t, v)) + b_{s,b} + \delta_s\big), \qquad (1)$$

where $b_{s,b}$ is a multicalibration offset and $\delta_s$ is a group-wise logit bias produced by a PI primal–dual controller. Both offsets are clipped for numerical stability (see code defaults).

**Multicalibration update (calibrating $r^{(0)}$).** The multicalibrator targets the no-exposure residual $r^{(0)} = Y^{(0)} - \hat{p}$. Each round, we use the most recent DR residual buffer to estimate slice-wise mean residuals $\widehat{\mathbb{E}}[r^{(0)} \mid A = s, \hat{p} \in I]$, select an active set of the most violated slices (budget $B_{\text{act}}$), and update the corresponding offsets $b_{s,b}$ for buckets $I = I_{s,b}$ with a clipped step. Importantly, this calibrator is applied in deployment phases (and optionally in pre-phase).

**PI primal–dual controller (auditing and steering $r^{(\Delta)}$).** To control treatment-effect parity and anti-rationing, COPF audits the *effect residual* $r^{(\Delta)} = (Y^{(1)} - Y^{(0)}) - \tau(W_t)$, where $\tau(\cdot)$ is the centering target for effect auditing. For the residual-only treatment-effect certificate, we use the group-mean-invariant centering in Lemma 4.1, e.g., a scalar $\bar{\tau}$ held fixed within the current audit window; the controller still estimates the group effects $\hat{\tau}_s$ separately from GA-DR pseudo-outcomes. Each round, we compute windowed gaps $(g_{\text{gap}}^{\text{TE}}, g_{\text{max}}^{\text{Cal}}, g^{\text{Min}})$ on the last $L_{\text{win}}$ samples. The PI controller maintains dual variables $(\lambda_{\text{TE}}, \lambda_{\text{Cal}}, \lambda_{\text{Min}})$ and updates them from soft violations (e.g., $v_{\text{TE}} = [g_{\text{gap}}^{\text{TE}} - \rho_{\text{TE}}]_+$, where $\rho_{\text{TE}}$ is the TE tolerance). The duals are then converted into per-group logit offsets using the current DR effect estimates $\hat{\tau}_s$:

$$\delta_s = \text{clip}\Big(\underbrace{\alpha\,\lambda_{\text{TE}}\,(\bar{\tau} - \hat{\tau}_s)}_{\text{TE parity}} + \underbrace{\alpha'\,\lambda_{\text{Min}}\,[\tau_{\min} - \hat{\tau}_s]_+}_{\text{minimum-effect guard}}\Big),$$

$$(2)$$

where $\bar{\tau}$ is the mean effect across groups. By default, we only apply these offsets in deployment and post-deployment phases, so the pre-phase serves as a warm-up. The controller also computes $\lambda_{\text{Cal}}$ with hierarchical gating (tighten calibration only after TE/Min are acceptable); in the released code, this coupling is optional and disabled by default.

---

**Algorithm 1** OPP Runner with COPF Modules

**Require:** Stream $\{(u_t, v_t^\star, t)\}_{t=1}^T$; backbone $f$; schedules $(K, \epsilon_q, \text{temp}_q)$; negative budget $N$; audit/log periods; window length $L_{\text{win}}$.
1: **for** $t = 1, \dots, T$ **do**
2:     Set phase $q_t \in \{\text{Pre}, \text{Deploy}, \text{Post}\}$ and build $C_t = \{v_t^\star\} \cup N_t, |N_t| \leq N$, from $G_{\leq t}$.
3:     Score all $v \in C_t$, assign group/slice labels, and form $\tilde{p}_t(v)$ by Eq. (1), with inactive modules setting $b_{s,b}$ or $\delta_s$ to zero.
4:     Sample $D_t$ using TOPK-STOCHASTIC with exploration $\epsilon_{q_t}$, temperature $\text{temp}_{q_t}$, and coverage modifiers when enabled; log $\hat{e}_t(v)$ for all $v \in C_t$.
5:     Observe $Y_t(v) = \mathbf{1}\{v = v_t^\star\}\mathbf{1}\{v \in D_t\}$; then update graph/backbone using positives.
6:     Update GA-DR buffers, cross-fit nuisance models, and pseudo-outcomes $\tilde{\Gamma}_t^{(0)}, \tilde{\Gamma}_t^{(1)}$.
7:     At audit checkpoints, update active-slice multicalibration offsets from $\hat{r}^{(0)}$; at log/control checkpoints, compute windowed gaps/certificates and update PI duals.
8: **end for**

---

**Online runner.** Algorithm 1 gives the prequential runner used by both Base and Base+COPF. The two share candidate construction, exposure logging, banditized feedback, and rolling-window evaluation; COPF enables the decision-layer offsets, auditors, and PI updates.

### 4.4. Deployment-Stability

We audit drift by running OPP under a base policy (**Pre**), changing exposure to induce performative shift (**Deploy**), then re-auditing on the evolved graph (**Post**).

*Remark* 4.7 (Complexity and coverage). Incremental neighborhood statistics + $k$-neighbor subsampling + $B_{\text{act}}$ active auditors yield amortized $O(|C_t| dL + |C_t|k + B_{\text{act}})$ per round for an $L$-layer model of width $d$. Coverage-driven exploration supports the group and slice masses required by Corollary 4.6.

## 5. Experiments

### 5.1. Datasets and Preprocessing

We evaluate COPF on two temporal interaction streams from the Temporal Graph Benchmark (TGB) (Huang et al., 2023): **tgbl-wiki** (user edits page), **tgbl-review** (user reviews item); and on a controlled synthetic bipartite stream. Each event $(u, v, t)$ is treated as one online round with source $u_t = u$ and true destination $v_t^\star = v$; the candidate set $C_t$ is formed online by adding up to $N$ negatives (Sec. 3).

**Synthetic protected attributes for TGB** The TGB streams do not provide demographic/sensitive attributes. To enable controlled evaluation *without claiming any real-world semantics*, we use synthetic binary group labels on nodes. Our default is a balanced ID-based split $A_v = \text{ID}(v) \bmod 2$, which serves as a "placebo" partition that stress-tests identifiability and the online auditing pipeline. To study a more structurally correlated partition, we also consider an activity/degree-based split in sensitivity experiments (Sec. 6). Unless stated otherwise, we audit groups on the destination side for TGB so that each candidate set spans both groups, making group-aware exposure control identifiable within each round.

**Synthetic bipartite stream** We generate a 600-user/4000-item stream with 200,000 events and injected group bias. Users carry the protected attribute $A \in \{0, 1\}$ and we audit on the user side (items have no group label).

## 5.2. Experimental Setup

**Evaluation protocol (Pre → Deploy → Post).** We follow a three-phase OPP protocol to measure deployment stability. Each phase contains 20,000 rounds (total 60,000): Pre (warm-up), Deploy (policy shift / performative feedback), and Post (stability under the shifted environment). Our exposure policy is TOPK-STOCHASTIC with $K = 10$ and a phase-dependent schedule: $\epsilon = 0.20$, temperature $= 1.0$ in Pre, and $\epsilon = 0.02$, temperature $= 0.7$ in Deploy/Post.

**Candidate generation and logged bandit feedback.** At round $t$ we treat the stream event $(u_t, v_t^\star)$ as the single positive and form the candidate set $C_t = \{v_t^\star\} \cup N_t$ by adding up to $N{=}200$ negatives sampled without replacement (item-only pool for the bipartite stream; all nodes for TGB). The exposure policy selects a slate $D_t$ of size $K$ and we log each candidate's marginal entry propensity $\hat{e}_t$. Feedback is bandit-style: we only observe whether the exposed slate contains the true destination, i.e., $Y_t(v) = \mathbf{1}\{v = v_t^\star\} D_t(v)$. This corresponds to a "platform-mediated" stress test with $Y_t^{(0)}(v) \equiv 0$ that isolates policy-induced selection: changing the exposure policy changes what can be observed and thus the future graph used for training and evaluation. COPF's definitions and estimators apply more generally when organic outcomes $Y^{(0)} \neq 0$; such counterfactual logs are rarely available in public benchmarks. Accordingly, in our runner the evolving graph (and any online backbone updates) are driven only by realized positives.

**Metrics and aggregation.** Utility is reported using ranking metrics computed on each candidate set (MRR, Hits@10, NDCG@10, DeployHit@TopK). Fairness is measured via the gaps in Def. 3.3, computed from GA-DR pseudo-outcomes on the same rolling audit window as

Residual-OI. For temporal plots we compute $g_{\text{gap}}^{\text{TE}}$ and $g_{\text{max}}^{\text{Cal}}$ every 1,000 rounds on the most recent $L_{\text{win}}$ logged candidates.

For phase-wise tables, we follow the aggregation protocol used by our runner logs. Within each seed and each phase, we (i) average checkpointed values inside that phase to obtain a *phase mean*, and (ii) compute a *worst-case-in-phase* summary (for utility we take the *minimum* within the phase; for fairness we take the *maximum* within the phase). We then report mean±std of the phase means across seeds, and in parentheses the average of the worst-case-in-phase summaries across seeds. Boldface indicates that a COPF-enabled method is better than its corresponding non-COPF counterpart on that statistic (higher for utility, lower for fairness). We also report the corresponding Residual-OI transfer certificates produced from the same audit window. All results are repeated over three random seeds; full phase-wise summaries are reported in Appendix Tables 4–6.

## 5.3. Baselines and Models

COPF is designed as a *decision-layer wrapper* around an arbitrary backbone scorer. We therefore evaluate COPF on three representative backbones commonly used for temporal link prediction: **EdgeBank** (Poursafaei et al., 2022), **TGN** (Rossi et al., 2020), and **GraphMixer** (Cong et al., 2023).

For *training-time* fairness baselines we implement three standard interventions for **TGN** only: **TGN-Adv** (adversarial debiasing), **TGN-Reweight** (group reweighting), and **TGN-Penalty** (fairness regularization). These baselines require differentiable end-to-end training and are not directly applicable to non-parametric EdgeBank; for GraphMixer we focus on COPF-as-wrapper comparisons.

Unless stated otherwise, we use TGN embedding/message dimensions of 128 and GraphMixer token/time-feature dimensions of 64 with two mixer layers and a time-gap of 2000.

## 5.4. Implementation Details

All experiments use the same OPP runner with Top-$K$ stochastic exposure: with probability $\epsilon$ we sample a size-$K$ slate uniformly without replacement; otherwise we sample from backbone scores with temperature. We log per-candidate marginal entry propensities: uniform exploration has analytic inclusion probabilities, while the Plackett–Luce Top-$K$ component uses Monte Carlo inclusion estimates (default 128 samples), and we clip the logged propensities for numerical stability. We also clip backbone probabilities to $[10^{-4}, 1 - 10^{-4}]$. Counterfactual metrics are estimated online with GA-DR plug-in estimators and lightweight nuisance models, using stabilized graph-aware time-decay

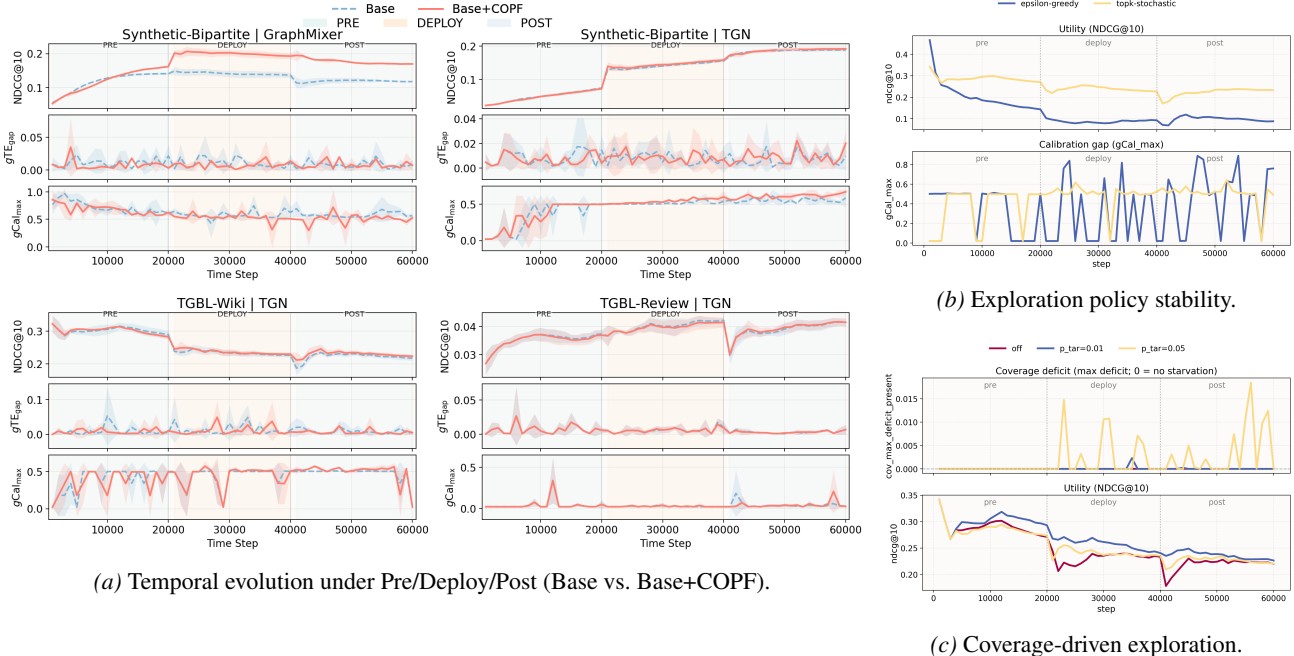

*(a)* Temporal evolution under Pre/Deploy/Post (Base vs. Base+COPF).

*(b)* Exploration policy stability.

*(c)* Coverage-driven exploration.

*Figure 1.* **Temporal evolution and key policy ablations.** (a) Pre/Deploy/Post trajectories for Base vs. Base+COPF, reporting NDCG@10 and counterfactual gaps ($g_{\text{gap}}^{\text{TE}}$, $g_{\text{max}}^{\text{Cal}}$) as mean $\pm 1$ std across seeds. (b) Exploration-policy stability comparing $\epsilon$-greedy and TOPK-STOCHASTIC at the same exploration rate. (c) Coverage-driven exploration as the minimum exposure target $p_{\text{tar}}$ varies. Vertical dashed lines mark phase boundaries.

weights. Auditing/multicalibration use group×score buckets (10 per group) with a bounded active set; richer slice families are supported but off unless stated. We log metrics every 1000 rounds. Dataset-specific hyperparameters are in Table 3; for Base models we disable auditing. Experiments run on a Linux server with GeForce RTX 4090 GPUs (24 GB); each run uses a single GPU for the backbone, while auditing and nuisance updates run on CPU.

## 6. Results

**Phase-wise performance and deployment stability** Appendix Tables 4–6 summarize utility (NDCG@10) and counterfactual gaps ($g_{\text{gap}}^{\text{TE}}$, $g_{\text{max}}^{\text{Cal}}$) under the Pre→Deploy→Post protocol. Across datasets, COPF behaves as a deployment-time decision layer rather than a retrained backbone: it leaves the policy nearly unchanged when the logged counterfactual gaps are already small, but becomes more active when the exposure shift creates unstable treatment effects. The clearest case is the injected-bias synthetic stream with GraphMixer (Table 5). During Deploy, Base+COPF improves NDCG@10 from 0.1417 to 0.1996 and Hits@10 from 0.2952 to 0.4154, while reducing mean $g_{\text{gap}}^{\text{TE}}$ from 0.0103 to 0.0076 and mean $g_{\text{max}}^{\text{Cal}}$ from 0.5816 to 0.5337. The worst-case-in-phase statistics show the same stabilization: Deploy $g_{\text{gap}}^{\text{TE}}$ decreases from 0.0477 to 0.0274 and $g_{\text{max}}^{\text{Cal}}$ from 0.9178 to 0.7067. In Post, the utility gain per-

sists ($0.1193 \rightarrow 0.1768$ NDCG@10), with lower calibration error ($0.5867 \rightarrow 0.5076$) and a slightly smaller treatment-effect gap ($0.0080 \rightarrow 0.0075$). By contrast, on EdgeBank and some TGN settings the changes are small or mixed, indicating that COPF is most useful when the backbone and policy shift create exploitable exposure-side instability.

On the real TGB streams, the gains are more localized, which is expected because the default ID-mod2 partition is a placebo split rather than a demographic attribute. On Wiki with TGN (Table 4), COPF reduces the PRE worst-case $g_{\text{gap}}^{\text{TE}}$ from 0.0528 to 0.0217, while keeping phase-mean utility essentially stable. After deployment, it slightly improves NDCG@10 in both Deploy ($0.2335 \rightarrow 0.2365$) and Post ($0.2219 \rightarrow 0.2296$), and reduces the POST mean $g_{\text{gap}}^{\text{TE}}$ from 0.0090 to 0.0067. The calibration diagnostic is less uniform: for example, Wiki+TGN POST $g_{\text{max}}^{\text{Cal}}$ increases from 0.4727 to 0.4892. We therefore treat $g_{\text{max}}^{\text{Cal}}$ as a high-variance diagnostic of counterfactual calibration under banditized feedback, while $g_{\text{gap}}^{\text{TE}}$ is the primary benefit-parity target. On Review (Table 6), treatment-effect gaps are already small for TGN, and COPF is correspondingly non-binding: POST NDCG@10 changes from 0.0389 to 0.0392, while POST $g_{\text{gap}}^{\text{TE}}$ changes from 0.0030 to 0.0029. The TGN training-time debiasing variants show similar behavior under the shared OPP pipeline, suggesting that COPF is best interpreted as an orthogonal deployment-time wrapper rather than a replacement for representation-level debiasing.

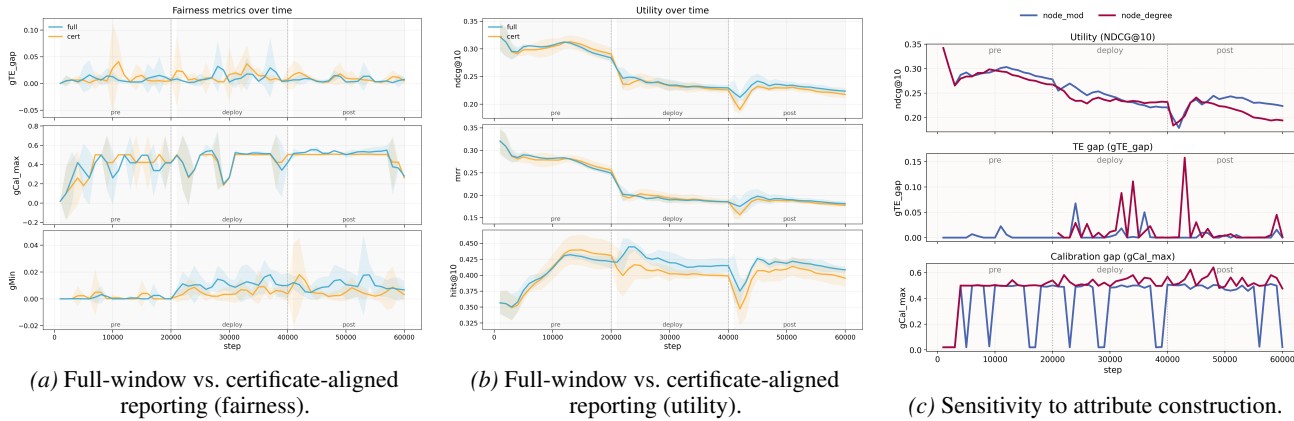

*(a)* Full-window vs. certificate-aligned reporting (fairness).

*(b)* Full-window vs. certificate-aligned reporting (utility).

*(c)* Sensitivity to attribute construction.

*Figure 2.* **Certificate alignment and sensitivity.** (a) Full-window vs. certificate-aligned reporting using the Residual-OI slice set for fairness metrics ($g_{\mathrm{gap}}^{\mathrm{TE}}$, $g_{\mathrm{max}}^{\mathrm{Cal}}$, $g^{\mathrm{Min}}$). (b) The same comparison for utility metrics (NDCG@10, MRR, Hits@10). (c) Sensitivity to TGB protected-attribute construction, comparing ID-mod2 with degree/activity splits. Shaded bands denote $\pm 1$ std across seeds; vertical dashed lines mark phase boundaries.

**Temporal evolution under policy shifts**  Figure 1a shows that the phase-wise averages hide short-lived transients around deployment boundaries. On Synthetic-Bipartite with GraphMixer, the utility curves separate after deployment in favor of Base+COPF, while the treatment-effect trajectory remains no worse and the calibration trajectory is smoother. On Wiki and Review, the mean TE gaps are small, but the rolling curves reveal occasional spikes and high-frequency calibration variation. COPF mainly suppresses these local excursions on Wiki and is almost indistinguishable from Base on Review, matching the table-level conclusion that the controller is active only when the logged counterfactual estimates indicate a violation. This is the intended deployment-stability behavior: the method does not force a uniform utility–fairness tradeoff in every phase, but monitors and corrects exposure-induced drift when it appears.

**Ablations and sensitivity**  The ablations support the same interpretation. Certificate-aligned reporting in Figures 2a and 2b closely tracks the full-window fairness and utility curves, indicating that the main trends are not artifacts of evaluating on a population different from the one used by the Residual-OI certificate. At the same time, the certificates remain conservative, especially when slice mass is small, so we use them as safety monitors rather than tight point estimates. The policy ablations in Figures 1b and 1c show that TOPK-STOCHASTIC exploration is smoother than $\epsilon$-greedy at the same exploration rate, and that modest coverage targets reduce short-term slice starvation without visibly dominating the ranking signal. Finally, the attribute-construction sensitivity in Figure 2c shows that replacing the placebo ID-mod2 split with a degree/activity-based split changes the baseline level of noise but not the qualitative conclusion: COPF is non-binding when counterfactual gaps are small and more active when exposure dynamics create instability.

The spike and certificate-slack diagnostics in Appendix H further reinforce this picture: Deploy spikes decrease and slack tightens on Synthetic-Bipartite with GraphMixer, Wiki with EdgeBank is nearly unchanged, and Wiki with TGN shows reduced PRE spikes with mixed later-phase behavior.

## 7. Conclusion and Future Work

We proposed **COPF**, a decision-layer framework for deployment-stable counterfactual fairness in online link recommendation on evolving graphs. COPF defines exposure-counterfactual group disparities with a minimum-effect guardrail, makes them identifiable from a single propensity-logged stream via explicit exploration, and audits/controls them via Residual-OI on GA-DR residuals. Our transfer analysis shows that plug-in Residual-OI implies high-probability bounds on true counterfactual gaps (with calibration reported as a diagnostic) under temporal mixing and bounded local interference. Empirically, COPF is largely non-binding when gaps are small, but under Pre/Deploy/Post policy shifts it reduces worst-case disparity spikes with modest utility impact.

At the same time, our public-benchmark evaluation is necessarily limited. TGB streams lack demographic attributes (so we use synthetic partitions), and our banditized outcome model stress-tests selection effects but makes calibration harder to interpret. Future work should strengthen the interference model beyond bounded local neighborhoods, improve propensity logging for complex slate policies, analyze convergence of budgeted active-auditor implementations under dependence, and expand comparisons to simpler online fairness controllers as well as settings with real sensitive attributes.

## Acknowledgements

Part of this work was completed by Sheng'en Li while participating in the summer and semester research programs at Duke University. We thank Professor Fan Wei of Duke University for helpful feedback and valuable suggestions that improved the paper. We also thank the anonymous reviewers for their insightful comments and constructive feedback.

## Impact Statement

This work develops methods for monitoring and controlling counterfactual fairness in online link recommendation. Its potential benefit is to make exposure policies more auditable under deployment feedback. Its risks include misuse of fairness certificates when assumptions, protected-attribute handling, or propensity logging are invalid. Real deployments should pair such certificates with domain validation, responsible data governance, and human oversight.

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

# Appendix

**Appendix roadmap.** Appendix A summarizes notation. Appendix B gives a schematic view of the OPP/COPF dataflow. Appendix C states checkable versions of the temporal mixing and bounded local interference conditions. Appendix D details the windowed Residual-OI / multicalibration routine and the finite-auditor confidence radius used by the logged certificates. Appendix E contains proofs for the theoretical results in Sec. 4.2. Appendix F summarizes implementation details and hyperparameters. Appendix G reports full phase-wise summary tables. Appendix H reports phase-wise spike and certificate-slack diagnostics. Appendix I reports propensity-logging sensitivity under IPS and GA-DR probes. Appendix J gives a target-pair feasibility diagnostic for the treatment-effect parity and minimum-effect guardrail constraints.

## A. Core Notation

For convenience, Table 1 collects the main symbols used throughout the paper (mainly Secs. 3, 4).

*Table 1.* Core notation.

| Symbol | Description |
|---|---|
| $t = 1, \dots, T$ | Online round; $T$ is the total horizon. |
| $G_{\leq t} = (V, E_{\leq t})$ | Pre-decision graph at round $t$. |
| $u_t, v_t^\star$ | Source node and stream destination revealed at round $t$. |
| $N_t, N$ | Online negative set and negative budget. |
| $C_t = \{v_t^\star\} \cup N_t$ | Candidate destination set; audited dyads are $\{(u_t, v) : v \in C_t\}$. |
| $K$ | Slate size. |
| $H_t$ | Policy history/decision information available before choosing the slate. |
| $\pi_t$ | Stochastic exposure policy. |
| $D_t \subseteq C_t, D_t(v)$ | Exposed slate and inclusion indicator $D_t(v) = \mathbf{1}\{v \in D_t\}$. |
| $e_t(v), \hat{e}_t(v)$ | True and logged marginal inclusion propensities. |
| $\hat{e}_t^{(1)}(v), \hat{e}_t^{(0)}(v)$ | Arm propensities: $\hat{e}_t(v)$ and $1 - \hat{e}_t(v)$. |
| $\epsilon$ | Exploration probability in the exposure policy. |
| $\phi_t(u_t, v), G_t^{\text{loc}}(u_t, v)$ | Dyad features and local graph summary. |
| $W_t(u_t, v)$ | Pre-exposure decision/counterfactual context; includes dyad features, local graph summary, and slate-policy inputs used by $\pi_t$. |
| $\hat{p}_t(u_t, v)$ | Backbone score/probability before COPF offsets. |
| $Y_t(u_t, v)$ | Observed outcome under the logged exposure. |
| $Y_t^{(0)}, Y_t^{(1)}$ | No-exposure and exposure potential outcomes. |
| $A_i, \mathcal{A}$ | Node protected attribute and finite group set. |
| $A_t(u_t, v), s$ | Dyad group label and a generic group $s \in \mathcal{A}$. |
| $B_{\text{bucket}}, I_{s,b}$ | Number of score buckets and bucket $b$ for group $s$. |
| $R_t(u_t, v), \mathcal{R}, \rho$ | Optional structural role, role set, and role value. |
| $h_{s,b,\rho}, \mathcal{H}$ | Slice indicator and auditor family. |
| $w_t$ | Nonnegative graph-aware weight. |
| $\mathcal{W}_k, L_{\text{win}}$ | Audit window and nominal window length. |
| $\widehat{\mathbb{E}}_{\text{GA}, \mathcal{W}_k}$ | Self-normalized GA empirical average over $\mathcal{W}_k$. |
| $W_{\text{eff}}$ | Effective sample size of a weighted window. |
| $\mu_a, \hat{\mu}_{a,t}$ | Arm-$a$ outcome regression and online nuisance estimate. |
| $\tilde{\Gamma}_t^{(a)}$ | GA-DR pseudo-outcome for arm $a$. |
| $r^{(0)}, r^{(\Delta)}$ | No-exposure and treatment-effect residuals. |
| $\tau_s, \hat{\tau}_s$ | True and estimated group exposure treatment effect. |
| $\tau(W_t), \bar{\tau}, \tau_{\min}$ | Effect-centering target, scalar window centering, and minimum-effect target. |
| $g_{\text{gap}}^{\text{TE}}$ | Treatment-effect parity gap. |
| $g_{\max}^{\text{Cal}}$ | Maximum within-group counterfactual calibration diagnostic. |
| $g^{\text{Min}}$ | Minimum-effect guardrail violation. |
| $p_{\min}^{\text{g}}, p_{\min}^{\text{gb}}$ | Minimum group mass and group-bucket slice mass. |
| $\varepsilon_T, \beta_T, E_T$ | Residual-OI tolerance, concentration radius, and total transfer error. |
| $\varepsilon_e, \varepsilon_\mu$ | Propensity and outcome-nuisance errors. |
| $\tau_{\text{mix}}, \kappa$ | Mixing scale and local dependency-graph degree. |
| $\tilde{p}_t, b_{s,b}, \delta_s$ | COPF-adjusted probability, calibration offset, and group logit offset. |
| $\lambda_{\text{TE}}, \lambda_{\text{Cal}}, \lambda_{\text{Min}}, \rho_{\text{TE}}$ | Controller dual variables and TE target tolerance. |

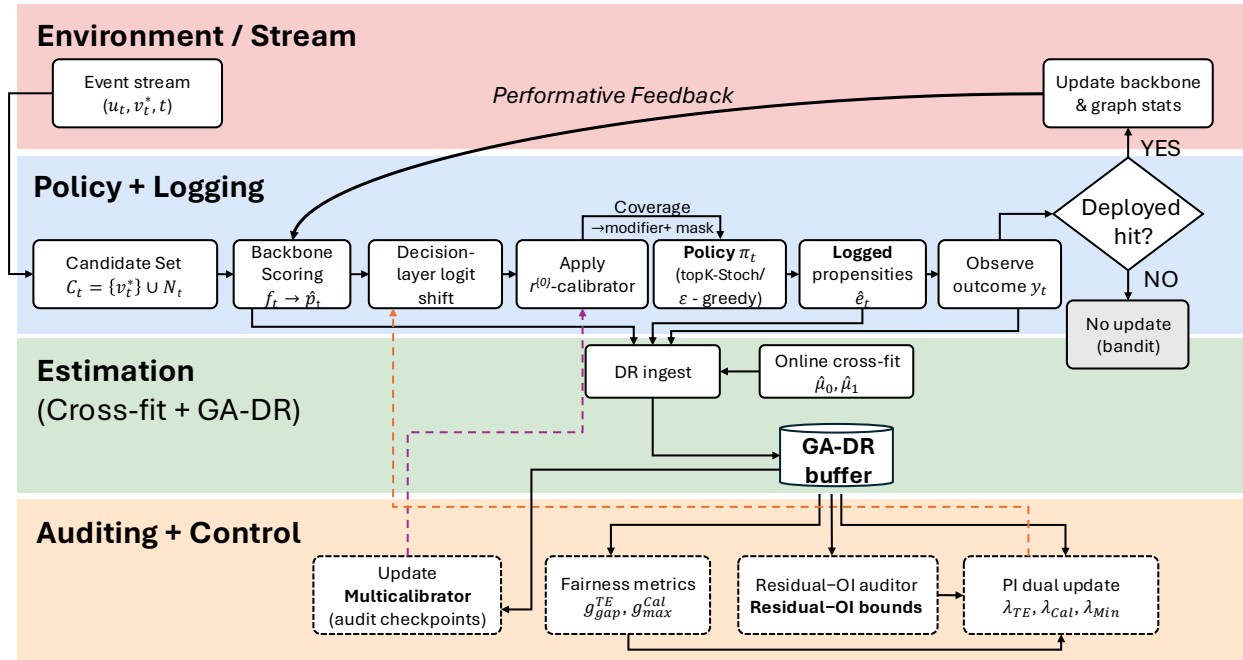

*Figure 3.* **Prequential logging protocol (OPP).** At each round we build the candidate set online, apply a stochastic Top-$K$ exposure policy with explicit exploration and propensity logging, observe banditized feedback on exposed candidates, and update the backbone and nuisance models sequentially. All reported counterfactual metrics and certificates are computed prequentially from the logged stream.

## B. Protocol

Figure 3 gives a schematic view of the OPP/COPF dataflow corresponding to Algorithm 1. Candidates are constructed prospectively, exposure is stochastic with explicit exploration, per-candidate marginal propensities are logged, and the same rolling log feeds GA-DR estimation, Residual-OI certification, multicalibration, and PI control.

## C. Checkable Definitions for Mixing and Local Interference

This appendix states operational (i.e., checkable) versions of the temporal dependence and interference conditions used throughout the paper. These conditions instantiate the bounded-dependence clause in Assumption 3.1 and justify the dependence factors in our transfer bounds. Let $Z_t$ denote the logged tuple(s) at time $t$ (e.g., context $W_t$, decisions $D_t$, propensities $\hat{e}_t$, outcomes $Y_t$, and any derived quantities such as DR pseudo-outcomes/residuals), and let $\mathcal{F}_t = \sigma(Z_1, \ldots, Z_t)$ be the natural filtration.

### C.1. Temporal mixing

**Definition C.1** ($\beta$-mixing (absolute regularity)). Define the $\beta$-mixing coefficients

$$\beta(\ell) \;\triangleq\; \sup_{t \geq 1} \; \mathbb{E}\left[\sup_{B \in \sigma(Z_{t+\ell}, Z_{t+\ell+1}, \ldots)} \left|\mathbb{P}(B \mid \mathcal{F}_t) - \mathbb{P}(B)\right|\right].$$

We say the process is $\beta$-mixing with mixing time $\tau_{\mathrm{mix}}$ (at tolerance $\varepsilon_{\mathrm{mix}}$) if $\beta(\tau_{\mathrm{mix}}) \leq \varepsilon_{\mathrm{mix}}$ (or more generally if $\beta(\ell)$ decays with $\ell$).

**Definition C.2** (Function-class mixing (finite test class)). Let $\mathcal{G}$ be a *finite* class of bounded test functions $g : \mathsf{Z} \to [-1, 1]$. Define the $\mathcal{G}$-mixing coefficient at lag $\ell$ by

$$\alpha_{\mathcal{G}}(\ell) \;\triangleq\; \sup_{t \geq 1} \; \sup_{g \in \mathcal{G}} \left|\mathrm{Cov}(g(Z_t), g(Z_{t+\ell}))\right|.$$

We say the process is $(\tau_{\mathrm{mix}}, \varepsilon_{\mathrm{mix}})$-$\mathcal{G}$-*mixing* if $\alpha_{\mathcal{G}}(\ell) \leq \varepsilon_{\mathrm{mix}}$ for all $\ell \geq \tau_{\mathrm{mix}}$.

*Remark* C.3 (Function-class mixing as a diagnostic). When $\mathcal{G}$ is finite (as in our auditing setting, e.g., slice indicators and their products with clipped residuals), $\alpha_{\mathcal{G}}(\ell)$ can be estimated by empirical lag-$\ell$ covariances computed on logged streams, optionally using a block bootstrap to attach uncertainty. In practice we only require $\mathcal{G}$ large enough to cover the statistics that enter the certificate, e.g., $g_t = h(W_t)\,\hat{r}_t$ for $h \in \mathcal{H}$ and bounded residual estimates $\hat{r}_t$. This function-class coefficient is used as an empirical diagnostic; the finite-sample concentration in Appendix D relies on the beta-mixing/blocking and dependency-graph conditions.

### C.2. Bounded local interference

The local-interference mapping is used to control dependence of logged quantities; our estimand remains the own-exposure marginal $Y^{(a)}$, not an arbitrary joint intervention.

**Definition C.4** (Local interference neighborhood and exposure mapping). Index an audited instance by $u = (i, j, t)$ (a candidate dyad at time $t$). Let $\mathbf{D}$ denote the global treatment/exposure assignment over all instances, and let $\mathbf{D}_{\mathcal{I}(u)}$ be the restriction to a subset $\mathcal{I}(u)$. We say the deployment satisfies $(r_{\mathrm{int}}, L_{\mathrm{int}})$-*local interference* if there exists: (i) a *computable* interference neighborhood $\mathcal{I}(u)$ computed from $G_{\leq t}$ and the candidate construction rule, such that $\mathcal{I}(u)$ contains only instances within $r_{\mathrm{int}}$ hops of $i$ or $j$ in $G_{\leq t}$ and within the last $L_{\mathrm{int}}$ rounds; and (ii) an exposure mapping $g_u(\cdot)$ such that the potential outcomes satisfy

$$Y_u(\mathbf{D}) = Y_u\big(g_u(\mathbf{D}_{\mathcal{I}(u)})\big),$$

i.e., treatments outside $\mathcal{I}(u)$ do not affect $Y_u$.

**Definition C.5** (Bounded-degree dependency graph). Define a (random) dependency graph on instances within a window: connect $u$ and $u'$ if

$$\mathcal{I}(u) \cap \mathcal{I}(u') \neq \emptyset,$$

or more generally if their logged tuples share any random variable entering the exposure mapping, nuisance estimates, or residuals. We assume the maximum degree is bounded by $\kappa$ (uniformly over time/windows). This is checkable because $\mathcal{I}(u)$ is computable from $G_{\leq t}$ and the candidate rule; $\kappa$ can be upper-bounded directly by design (e.g., $k$-hop subsampling, capped candidate budgets, bounded memory).

*Remark* C.6 (No-interference special case). No outcome interference is recovered by taking $r_{\mathrm{int}} = 0$ and $L_{\mathrm{int}} = 0$, so each potential outcome depends only on its own exposure. The dependency degree $\kappa$ may still include within-slate dependence induced by Top-$K$ sampling without replacement; it is zero only under independent candidate-level exposure.

## D. Windowed Residual-OI with Active Auditors

This appendix specifies the windowed Residual-OI / multicalibration subroutine used at audit checkpoints in Algorithm 1. The subroutine operates on a finite auditor class $\mathcal{H}$, maintains an active-auditor budget, and supplies the confidence radii used in the logged certificates.

### D.1. Setup and certificate

Fix a finite auditor family $\mathcal{H}$ of bounded functions $h : \mathsf{W} \to [-1, 1]$, where $\mathsf{W}$ is the logged context space (e.g., group indicators, score buckets, optional structural roles). Let $\hat{r}_t$ denote the (clipped) residual estimate available online, e.g., a GA-DR pseudo-outcome residual $\hat{r}_t^{(0)} = \tilde{\Gamma}_t^{(0)} - \hat{p}_t$ for calibration, or $\hat{r}_t^{(\Delta)} = (\tilde{\Gamma}_t^{(1)} - \tilde{\Gamma}_t^{(0)}) - \tau(W_t)$ for effect auditing.

We evaluate Residual-OI on a sliding window $\mathcal{W}_k$ of nominal length $L_{\mathrm{win}}$ using the same graph-aware self-normalized averaging operator as Def. 3.2:

$$\widehat{\mathbb{E}}_{\mathrm{GA}, \mathcal{W}_k}[Z] \triangleq \frac{\sum_{t \in \mathcal{W}_k} w_t\, Z_t}{\sum_{t \in \mathcal{W}_k} w_t},$$

$$\widehat{\mathrm{OI}}_{\mathrm{GA}}(\hat{r}; \mathcal{H}, \mathcal{W}_k) \triangleq \sup_{h \in \mathcal{H}} \left| \widehat{\mathbb{E}}_{\mathrm{GA}, \mathcal{W}_k}\big[h(W_t)\,\hat{r}_t\big] \right|.$$

(When $\mathcal{H}$ consists of slice indicators, this recovers windowed multicalibration.)

**Online-loop interface.** The complete OPP runner with COPF modules is stated in Algorithm 1. This appendix does not restate the full loop. Instead, it specifies the windowed Residual-OI / multicalibration subroutine invoked at audit checkpoints and the confidence radius used by the logged certificates.

## D.2. Algorithm: budgeted windowed Residual-OI (active auditors)

We maintain a sparse set of active auditors $\mathcal{H}^{\mathrm{act}}$ of size at most $B_{\mathrm{act}}$, together with per-auditor offsets (or weights) $b_h$ that define the current adjustment. For slice-based multicalibration, $h$ indexes a slice $S$ and $b_h$ corresponds to the slice offset applied in the decision layer (Eq. (1)).

---

**Algorithm 2** Budgeted windowed Residual-OI / multicalibration with active auditors.

---

**Input:** finite auditor class $\mathcal{H}$; window length $L_{\mathrm{win}}$; tolerance $\alpha$; active budget $B_{\mathrm{act}}$; step size $\eta$; offset radius $b_{\max}$; confidence radius $\beta_W(\delta)$ (Def. D.1).
**State:** active set $\mathcal{H}^{\mathrm{act}}$ and offsets $\{b_h\}_{h \in \mathcal{H}^{\mathrm{act}}}$; projection $\Pi_{[-b_{\max}, b_{\max}]}(\cdot)$.

1: **for** each window $\mathcal{W}_k$ (most recent $L_{\mathrm{win}}$ logged instances) **do**
2:     For each $h \in \mathcal{H}$ compute the window correlation $\hat{\rho}_k(h) = \widehat{\mathbb{E}}_{\mathrm{GA}, \mathcal{W}_k}[h(W_t)\,\hat{r}_t]$.
3:     Let $\mathcal{V}$ be the set of up to $B_{\mathrm{act}}$ auditors with the largest $|\hat{\rho}_k(h)|$ satisfying $|\hat{\rho}_k(h)| > \alpha + \beta_W(\delta)$.
4:     **for** each $h \in \mathcal{V}$ **do**
5:         Set $\mathcal{H}^{\mathrm{act}} = \mathcal{H}^{\mathrm{act}} \cup \{h\}$.
6:         Update its offset by projection: $b_h = \Pi_{[-b_{\max}, b_{\max}]}(b_h + \eta\,\hat{\rho}_k(h))$.
7:     **end for**
8:     (Optional) If $|\mathcal{H}^{\mathrm{act}}| > B_{\mathrm{act}}$, keep only the $B_{\mathrm{act}}$ active auditors with largest $|b_h|$.
9: **end for**

---

**Remarks.** (i) The routine only *updates* $B_{\mathrm{act}}$ auditors per window, but it may still *score* all $|\mathcal{H}|$ auditors to find the top violations. This matches the common "active constraints" implementation in COPF. (ii) For slice indicators, the correlation $\hat{\rho}_k(h)$ is proportional to the slice mean residual; one may equivalently compute slice means directly, with a minimum-mass threshold $p_{\min}$.

## D.3. Concentration radius for a finite auditor class

We state a standard deviation bound for the window correlations under the checkable dependence conditions of Appendix C. Because we use self-normalized GA weights, the natural effective sample size is

$$W_{\mathrm{eff}} \triangleq \frac{\left(\sum_{t \in \mathcal{W}_k} w_t\right)^2}{\sum_{t \in \mathcal{W}_k} w_t^2},$$

and one may replace $L_{\mathrm{win}}$ by $W_{\mathrm{eff}}$ in what follows (we do so in the logged implementation).

We assume the GA weights are nonnegative, uniformly bounded, and predictable with respect to the pre-outcome filtration; in particular, they may depend on $G_{\leq t}$, $W_t$, and time, but not on the unobserved potential outcomes or the post-outcome residual noise at $t$.

**Definition D.1** (Window confidence radius). For brevity, in this appendix $t$ indexes flattened logged candidate instances $(t, v)$ within the audit window.

Assume: (i) bounded residuals $|\hat{r}_t| \leq 1$ (achieved by clipping), (ii) a blocking/coupling condition, e.g. $\beta(\tau_{\mathrm{mix}}) \leq \varepsilon_{\mathrm{mix}}$, sufficient to reduce the window to approximately independent blocks, and (iii) a bounded-degree dependency graph of maximum degree $\kappa$ within each window (Def. C.5). Define

$$\beta_W(\delta) \triangleq c\sqrt{\frac{(\kappa + 1)\tau_{\mathrm{mix}}\left(\log|\mathcal{H}| + \log(1/\delta)\right)}{W_{\mathrm{eff}}}} + c'\varepsilon_{\mathrm{mix}}$$

where $c, c' > 0$ are universal constants.

**Lemma D.2** (Hoeffding inequality for dependency graphs). *Let $V_1, \ldots, V_n$ be real-valued random variables that admit a dependency graph of maximum degree $D$, and suppose almost surely $V_i \in [a_i, b_i]$. Then for any $t > 0$,*

$$\Pr\left(\sum_{i=1}^{n}(V_i - \mathbb{E}V_i) \geq t\right) \leq \exp\left(-\frac{2t^2}{(D + 1)\sum_{i=1}^{n}(b_i - a_i)^2}\right).$$

*The same bound holds for* $\Pr(\sum(V_i - \mathbb{E}V_i) \leq -t)$, *and hence*

$$\Pr\Big(\Big|\sum_{i=1}^{n}(V_i - \mathbb{E}V_i)\Big| \geq t\Big) \leq 2\exp\Big(-\frac{2t^2}{(D+1)\sum_{i=1}^{n}(b_i - a_i)^2}\Big).$$

*Proof.* Let $\chi$ denote the chromatic number of the dependency graph. A graph of maximum degree $D$ satisfies $\chi \leq D+1$ (greedy coloring). Let $\mathcal{I}_1, \ldots, \mathcal{I}_\chi$ be the color classes, so that within each class the variables are jointly independent. Define $S = \sum_{i=1}^{n}(V_i - \mathbb{E}V_i)$ and $S_j = \sum_{i \in \mathcal{I}_j}(V_i - \mathbb{E}V_i)$ so that $S = \sum_{j=1}^{\chi} S_j$.

Fix $\lambda > 0$. By Hölder's inequality with exponents $(\chi, \ldots, \chi)$,

$$\mathbb{E}\big[e^{\lambda S}\big] = \mathbb{E}\left[\prod_{j=1}^{\chi} e^{\lambda S_j}\right] \leq \prod_{j=1}^{\chi}\Big(\mathbb{E}\big[e^{\chi\lambda S_j}\big]\Big)^{1/\chi}.$$

For each $j$, the summands in $S_j$ are independent and bounded in intervals of width $(b_i - a_i)$. Hoeffding's lemma gives

$$\mathbb{E}\big[e^{\chi\lambda S_j}\big] \leq \exp\Big(\frac{(\chi\lambda)^2}{8}\sum_{i \in \mathcal{I}_j}(b_i - a_i)^2\Big).$$

Combining the above displays yields

$$\mathbb{E}\big[e^{\lambda S}\big] \leq \exp\Big(\frac{\chi\lambda^2}{8}\sum_{i=1}^{n}(b_i - a_i)^2\Big).$$

Applying Markov's inequality,

$$\Pr(S \geq t) \leq \exp(-\lambda t)\,\mathbb{E}[e^{\lambda S}]$$

$$\leq \exp\Big(-\lambda t + \frac{\chi\lambda^2}{8}\sum_{i=1}^{n}(b_i - a_i)^2\Big)$$

Optimizing over $\lambda$ gives $\lambda^\star = \frac{4t}{\chi\sum_i(b_i-a_i)^2}$ and hence

$$\Pr(S \geq t) \leq \exp\Big(-\frac{2t^2}{\chi\sum_{i=1}^{n}(b_i - a_i)^2}\Big)$$

$$\leq \exp\Big(-\frac{2t^2}{(D+1)\sum_{i=1}^{n}(b_i - a_i)^2}\Big),$$

since $\chi \leq D+1$. The lower-tail bound follows by applying the same argument to $-V_i$. $\qquad\square$

**Definition D.3** (Within-window GA population average). Fix a window $\mathcal{W}_k$ and let $\bar{w}_t = w_t/\sum_{s \in \mathcal{W}_k} w_s$ be the normalized GA weights. For predictable weights, define the within-window population GA average conditionally on the pre-outcome information:

$$\mathbb{E}_{\mathrm{GA}, \mathcal{W}_k}[Z] := \sum_{t \in \mathcal{W}_k} \bar{w}_t\,\mathbb{E}[Z_t \mid \mathcal{F}_t^-],$$

where $\mathcal{F}_t^-$ contains the realized contexts, weights, candidate sets, and policy information before the outcome noise at $t$.

**Theorem D.4** (Uniform deviation of windowed GA correlations). *Fix a window $\mathcal{W}_k$ and use the population GA average $\mathbb{E}_{\mathrm{GA}, \mathcal{W}_k}[\cdot]$ from Def. D.3.*

*Under the conditions of Def. D.1, with probability at least $1 - \delta$,*

$$\sup_{h \in \mathcal{H}}\Big|\widehat{\mathbb{E}}_{\mathrm{GA}, \mathcal{W}_k}\big[h(W_t)\,\hat{r}_t\big] - \mathbb{E}_{\mathrm{GA}, \mathcal{W}_k}\big[h(W_t)\,\hat{r}_t\big]\Big| \leq \beta_W(\delta).$$

*In particular, if the logged certificate satisfies $\widehat{\mathrm{OI}}_{\mathrm{GA}}(\hat{r}; \mathcal{H}, \mathcal{W}_k) \leq \alpha$, then the corresponding (within-window) population residual-OI gap obeys*

$$\sup_{h \in \mathcal{H}}\big|\mathbb{E}_{\mathrm{GA}, \mathcal{W}_k}\big[h(W_t)\,\hat{r}_t\big]\big| \leq \alpha + \beta_W(\delta).$$

*Proof.* Condition on the pre-outcome information in the window, so the realized weights and normalized weights $\bar{w}_t = w_t / \sum_{s \in \mathcal{W}_k} w_s$ are fixed and $\sum_{t \in \mathcal{W}_k} \bar{w}_t = 1$. For a fixed auditor $h \in \mathcal{H}$ define the bounded sequence

$$X_t(h) \triangleq h(W_t)\,\hat{r}_t \in [-1, 1],$$

$$V_t(h) \triangleq \bar{w}_t\, X_t(h) \in [-\bar{w}_t, \bar{w}_t],$$

and the centered sum $S(h) \triangleq \sum_{t \in \mathcal{W}_k} \big( V_t(h) - \mathbb{E}[V_t(h) \mid \mathcal{F}_t^-] \big)$. By definition, $\widehat{\mathbb{E}}_{\mathrm{GA}, \mathcal{W}_k}[X(h)] - \mathbb{E}_{\mathrm{GA}, \mathcal{W}_k}[X(h)] = S(h)$.

**Concentration for a fixed auditor.** Under Def. D.1, the windowed variables admit a dependency graph whose maximum degree is bounded (up to constants) by $D \lesssim \kappa\,\tau_{\mathrm{mix}}$; this captures (i) bounded local interference (degree $\kappa$) and (ii) short-range temporal dependence over $\tau_{\mathrm{mix}}$ lags. Applying Lemma D.2 with $V_i = V_t(h)$ gives, for any $u > 0$,

$$\Pr\big(|S(h)| \geq u\big) \leq 2\exp\Big( -\frac{2u^2}{(D+1)\sum_{t \in \mathcal{W}_k}(2\bar{w}_t)^2} \Big).$$

Since $\sum_{t \in \mathcal{W}_k}(2\bar{w}_t)^2 = 4\sum_{t \in \mathcal{W}_k} \bar{w}_t^2 = 4/W_{\mathrm{eff}}$, we obtain

$$\Pr\big(|S(h)| \geq u\big) \leq 2\exp\Big( -\frac{u^2 W_{\mathrm{eff}}}{2(D+1)} \Big).$$

**Union bound over $\mathcal{H}$.** Let $M = |\mathcal{H}| < \infty$. Choose $u = c\sqrt{\frac{(D+1)\big(\log M + \log(2/\delta)\big)}{W_{\mathrm{eff}}}}$ for a sufficiently large universal constant $c$. Then $2\exp\big( -\frac{u^2 W_{\mathrm{eff}}}{2(D+1)} \big) \leq \delta/M$, and a union bound yields

$$\Pr\Big( \sup_{h \in \mathcal{H}} |S(h)| \leq u \Big) \geq 1 - \delta.$$

Using $D + 1 \lesssim (\kappa+1)(\tau_{\mathrm{mix}}+1)$ and absorbing constant factors into $c$ gives the stated radius $c\sqrt{\frac{\kappa\,\tau_{\mathrm{mix}}(\log|\mathcal{H}| + \log(1/\delta))}{W_{\mathrm{eff}}}}$.

**Approximate mixing.** If the temporal dependence is only approximate beyond lag $\tau_{\mathrm{mix}}$ (e.g., $\beta(\tau_{\mathrm{mix}}) \leq \varepsilon_{\mathrm{mix}}$), a standard Berbee-type coupling argument yields the same bound with an additional $O(\varepsilon_{\mathrm{mix}})$ slack, which is recorded as the additive $c'\varepsilon_{\mathrm{mix}}$ term in Def. D.1. $\qquad\square$

**Corollary D.5** (Windowed multicalibration from residual-OI). *Let $\mathcal{H}$ contain slice indicators $h_S = \mathbf{1}\{(A, \hat{p}, \mathrm{role}) \in S\}$ and suppose each audited slice has within-window GA mass at least $p_{\min}$: $\mathbb{E}_{\mathrm{GA}, \mathcal{W}_k}[h_S(W_t)] \geq p_{\min}$. Then, with probability at least $1 - \delta$, simultaneously for all such slices,*

$$\Big| \mathbb{E}_{\mathrm{GA}, \mathcal{W}_k}[\hat{r}_t \mid S] \Big| \triangleq \left| \frac{\mathbb{E}_{\mathrm{GA}, \mathcal{W}_k}[h_S(W_t)\,\hat{r}_t]}{\mathbb{E}_{\mathrm{GA}, \mathcal{W}_k}[h_S(W_t)]} \right| \leq \frac{\alpha + \beta_W(\delta)}{p_{\min}}.$$

**Link to Assumption 4.3 and Thm. 4.5.** If we take a growing window length $L_{\mathrm{win}}$ and set $\alpha = 0$ (or $\alpha$ of order $L_{\mathrm{win}}^{-1/2}$), then Def. D.1 implies $\sup_{h \in \mathcal{H}} |\widehat{\mathbb{E}}[h\,\hat{r}] - \mathbb{E}[h\,\hat{r}]| = O\big(\sqrt{\kappa\tau_{\mathrm{mix}}\log|\mathcal{H}|/W_{\mathrm{eff}}}\big)$, matching the $W_{\mathrm{eff}}^{-1/2}$-type behavior in Assumption 4.3 (up to dependence factors). Thm. 4.5 then transfers this plug-in residual control from $\hat{r}$ to the true residual $r$, adding the nuisance-estimation and mixing terms.

**Implementation note: active-auditor budget.** In slice-based settings, only a small fraction of slices typically violate the tolerance at any given window. Updating only the top-$B_{\mathrm{act}}$ violated auditors is therefore an efficient approximation to full multicalibration/Residual-OI, and is consistent with standard "active constraints" practice in online constrained optimization.

# E. Proofs supporting Sec. 4.2

**Overview.** This section provides full proofs for Lemma 4.1 (linearization to residual correlations), Lemma 4.2 (finite-sample control of GA-DR slice means), Theorem 4.5 (transfer from noisy plug-in residual-OI to control of the true residual), and Corollary 4.6 (residual control implies bounds on counterfactual fairness gaps).

**Proof of Lemma 4.1.** The identities hold under any fixed evaluation distribution; in the windowed certificates below, expectations are interpreted under the GA-weighted audit-window population distribution.

Recall the calibration residual

$$r^{(0)} \triangleq Y^{(0)} - \hat{p}$$

and the effect residual

$$r^{(\Delta)} \triangleq (Y^{(1)} - Y^{(0)}) - \tau(W_t).$$

For calibration, by the tower property,

$$\mathbb{E}[r^{(0)} \mid A = s, \hat{p} \in I] = \mathbb{E}[Y^{(0)} \mid A = s, \hat{p} \in I]$$
$$- \mathbb{E}[\hat{p} \mid A = s, \hat{p} \in I].$$

Multiplying by $\Pr(A = s, \hat{p} \in I)$ yields

$$\mathbb{E}[\mathbf{1}\{A = s, \hat{p} \in I\} r^{(0)}] = \Pr(A = s, \hat{p} \in I) \cdot \left( \mathbb{E}[Y^{(0)} \mid A = s, \hat{p} \in I] - \mathbb{E}[\hat{p} \mid A = s, \hat{p} \in I] \right),$$

and taking absolute values gives the stated expression for $g_{s,I}^{\mathrm{Cal}}$.

For treatment effects, note that for any group $s$ with $\Pr(A = s) > 0$,

$$\frac{\mathbb{E}[\mathbf{1}\{A = s\} r^{(\Delta)}]}{\Pr(A = s)} = \mathbb{E}[Y^{(1)} - Y^{(0)} \mid A = s] - \mathbb{E}[\tau(W_t) \mid A = s].$$

Let $m_s \triangleq \mathbb{E}[\tau(W_t) \mid A = s]$. Under the group-mean-invariant centering condition in Lemma 4.1, $m_s = m_\tau$ for every audited group $s$. This condition is satisfied, for example, by a scalar effect-centering target $\tau(W_t) \equiv \bar{\tau}$ held fixed within the audit window. Hence $m_{s_1} - m_{s_2} = 0$, and

$$\mathbb{E}[Y^{(1)} - Y^{(0)} \mid A = s_1] - \mathbb{E}[Y^{(1)} - Y^{(0)} \mid A = s_2] = \frac{\mathbb{E}[\mathbf{1}\{A = s_1\} r^{(\Delta)}]}{\Pr(A = s_1)} - \frac{\mathbb{E}[\mathbf{1}\{A = s_2\} r^{(\Delta)}]}{\Pr(A = s_2)}.$$

The stated formula for $g_{s_1,s_2}^{\mathrm{TE}}$ follows by taking absolute values.

Finally, for the minimum-effect violation,

$$g_s^{\mathrm{Min}} = \left[ \tau_{\min} - \tau_s \right]_+, \qquad \tau_s = \mathbb{E}[Y^{(1)} - Y^{(0)} \mid A = s].$$

Using $\tau_s = \mathbb{E}[r^{(\Delta)} \mid A = s] + \mathbb{E}[\tau(W_t) \mid A = s]$ gives

$$g_s^{\mathrm{Min}} = \left[ \tau_{\min} - \left\{ \frac{\mathbb{E}[\mathbf{1}\{A = s\} r^{(\Delta)}]}{\Pr(A = s)} + \mathbb{E}[\tau(W_t) \mid A = s] \right\} \right]_+,$$

which is the claimed expression.

**Proof of Lemma 4.2.** Fix an arm $a \in \{0, 1\}$ and a bounded slice indicator $h : \mathsf{W} \to [-1, 1]$. For concision we suppress the time index when it is clear.

Let $e_a(W) \triangleq \Pr(D = a \mid W)$ be the (true) propensity and $\mu_a(W) \triangleq \mathbb{E}[Y \mid D = a, W]$ the (true) outcome regression. Define the (clipped, cross-fitted) GA-DR pseudo-outcome

$$\tilde{\Gamma}^{(a)} = \hat{\mu}_a(W) + \frac{\mathbf{1}\{D = a\}}{\hat{e}_a(W)} (Y - \hat{\mu}_a(W)),$$

We decompose the estimation error into a stochastic term and a DR bias term:

$$\widehat{\mathbb{E}}_{\mathrm{GA},\mathcal{W}}[h\tilde{\Gamma}^{(a)}] - \mathbb{E}_{\mathrm{GA},\mathcal{W}}[hY^{(a)}] = \underbrace{\widehat{\mathbb{E}}_{\mathrm{GA},\mathcal{W}}[h\tilde{\Gamma}^{(a)}] - \mathbb{E}_{\mathrm{GA},\mathcal{W}}[h\tilde{\Gamma}^{(a)}]}_{\text{(I) empirical process term}} + \underbrace{\mathbb{E}_{\mathrm{GA},\mathcal{W}}[h\tilde{\Gamma}^{(a)}] - \mathbb{E}_{\mathrm{GA},\mathcal{W}}[hY^{(a)}]}_{\text{(II) DR bias term}}.$$

where $\mathbb{E}_{\mathrm{GA},\mathcal{W}}[\cdot]$ is the within-window population quantity in Def. D.3.

*(II) DR bias term.* Conditioning on the pre-outcome information and on $W$, the usual DR identity gives $\mathbb{E}[Y - \hat{\mu}_a(W) \mid D = a, W] = \mu_a(W) - \hat{\mu}_a(W)$, and hence

$$\mathbb{E}[\tilde{\Gamma}^{(a)} \mid W] = \hat{\mu}_a(W) + \frac{\mathbb{E}[\mathbf{1}\{D = a\}(Y - \hat{\mu}_a(W)) \mid W]}{\hat{e}_a(W)}$$

$$= \hat{\mu}_a(W) + \frac{e_a(W)\big(\mu_a(W) - \hat{\mu}_a(W)\big)}{\hat{e}_a(W)}$$

$$= \mu_a(W) + \Big(\frac{e_a(W)}{\hat{e}_a(W)} - 1\Big)\big(\mu_a(W) - \hat{\mu}_a(W)\big).$$

Under Assumption 3.1, $\mathbb{E}[Y^{(a)} \mid W] = \mu_a(W)$, so

$$\mathbb{E}[h(W)\tilde{\Gamma}^{(a)}] - \mathbb{E}[h(W)Y^{(a)}] = \mathbb{E}\left[h(W)\Big(\frac{e_a(W)}{\hat{e}_a(W)} - 1\Big)\big(\mu_a(W) - \hat{\mu}_a(W)\big)\right]$$

Using $|h| \le 1$, $\hat{e}_a(W) \ge \underline{e}$ (clipping/overlap), and the uniform nuisance error bounds $\|\hat{e}_a - e_a\|_\infty \le \varepsilon_e$ and $\|\hat{\mu}_a - \mu_a\|_\infty \le \varepsilon_\mu$, we obtain

$$\left|\mathbb{E}[h(W)\tilde{\Gamma}^{(a)}] - \mathbb{E}[h(W)Y^{(a)}]\right| \le \mathbb{E}\left[\left|\frac{e_a(W) - \hat{e}_a(W)}{\hat{e}_a(W)}\right| \cdot |\mu_a(W) - \hat{\mu}_a(W)|\right] \le \frac{\varepsilon_e \varepsilon_\mu}{\underline{e}}.$$

Averaging this bound with the normalized GA weights over $\mathcal{W}$, and using $ab \le \frac{1}{2}(a^2 + b^2)$, gives the recorded $C_1 \varepsilon_e \varepsilon_\mu + C_2(\varepsilon_e^2 + \varepsilon_\mu^2)$ term.

*(I) Empirical process term.* By clipping, $\tilde{\Gamma}^{(a)}$ is uniformly bounded, so the sequence $Z_t \triangleq h(W_t)\tilde{\Gamma}_t^{(a)}$ is bounded. Applying Theorem D.4 uniformly over the finite auditor class $\mathcal{H}$ yields a deviation term of the form $C_3 \sqrt{\tau_{\mathrm{mix}} \log |\mathcal{H}|/W_{\mathrm{eff}}}$ (absorbing weights/effective sample size and dependence factors into constants), which completes the proof.

**Proof of Theorem 4.5.** Fix $h \in \mathcal{H}$. By adding and subtracting the plug-in empirical certificate and the plug-in population average,

$$|\mathbb{E}_{\mathrm{GA},\mathcal{W}}[h(W_t)r_t]| \le \left|\widehat{\mathbb{E}}_{\mathrm{GA},\mathcal{W}}[h(W_t)\hat{r}_t]\right|$$

$$+ \left|\mathbb{E}_{\mathrm{GA},\mathcal{W}}[h(W_t)\hat{r}_t] - \widehat{\mathbb{E}}_{\mathrm{GA},\mathcal{W}}[h(W_t)\hat{r}_t]\right|$$

$$+ |\mathbb{E}_{\mathrm{GA},\mathcal{W}}[h(W_t)(r_t - \hat{r}_t)]|.$$

Taking the supremum over $h \in \mathcal{H}$, the first term is bounded by $\varepsilon_T$ by the logged certificate, and the second by $c_4\beta_T$ via Theorem D.4. For the third term, consider the two residuals separately. If $\hat{r} = \hat{r}^{(0)}$, then $\hat{r}^{(0)} - r^{(0)} = \tilde{\Gamma}^{(0)} - Y^{(0)}$. If $\hat{r} = \hat{r}^{(\Delta)}$, then

$$\hat{r}^{(\Delta)} - r^{(\Delta)} = (\tilde{\Gamma}^{(1)} - Y^{(1)}) - (\tilde{\Gamma}^{(0)} - Y^{(0)}).$$

Applying Lemma 4.2 to arm 0, and to both arms 0, 1 in the effect case, gives

$$\sup_{h \in \mathcal{H}} |\mathbb{E}_{\mathrm{GA},\mathcal{W}}[h(W_t)(r_t - \hat{r}_t)]| \le c_1 \varepsilon_e \varepsilon_\mu + c_2(\varepsilon_e^2 + \varepsilon_\mu^2) + c_3\sqrt{\frac{\tau_{\mathrm{mix}} \log |\mathcal{H}|}{W_{\mathrm{eff}}}},$$

up to universal constants. Combining the three bounds proves the theorem.

**Proof of Corollary 4.6.** In this proof, $\mathbb{E}$ and $\Pr$ denote the GA-weighted audit-window population distribution. Let $R_T = \varepsilon_T + E_T$. By Theorem 4.5, with high probability,

$$\sup_{h \in \mathcal{H}} |\mathbb{E}[hr]| \le R_T$$

for the relevant true residual sequence $r$.

For calibration, take $h = \mathbf{1}\{A = s, \hat{p} \in I\}$. Since the audited group-bucket mass is at least $p_{\min}^{\text{gb}}$, Lemma 4.1 gives

$$g_{s,I}^{\text{Cal}} = \frac{\left|\mathbb{E}[\mathbf{1}\{A = s, \hat{p} \in I\}\, r^{(0)}]\right|}{\Pr(A = s, \hat{p} \in I)} \leq \frac{R_T}{p_{\min}^{\text{gb}}}.$$

For treatment-effect parity, take $h = \mathbf{1}\{A = s\}$. Under the group-mean-invariant effect centering in Lemma 4.1,

$$g_{s_1,s_2}^{\text{TE}} \leq \frac{\left|\mathbb{E}[\mathbf{1}\{A = s_1\}r^{(\Delta)}]\right|}{\Pr(A = s_1)} + \frac{\left|\mathbb{E}[\mathbf{1}\{A = s_2\}r^{(\Delta)}]\right|}{\Pr(A = s_2)} \leq \frac{2R_T}{p_{\min}^{\text{g}}}.$$

For the minimum-effect guard, Lemma 4.1 gives

$$g_s^{\text{Min}} = \left[\tau_{\min} - \left\{\frac{\mathbb{E}[\mathbf{1}\{A = s\}r^{(\Delta)}]}{\Pr(A = s)} + m_s\right\}\right]_+, \qquad m_s = \mathbb{E}[\tau(W_t) \mid A = s].$$

Since

$$\left|\frac{\mathbb{E}[\mathbf{1}\{A = s\}r^{(\Delta)}]}{\Pr(A = s)}\right| \leq \frac{R_T}{p_{\min}^{\text{g}}},$$

monotonicity of the hinge yields

$$g_s^{\text{Min}} \leq \left[\tau_{\min} - m_s + \frac{R_T}{p_{\min}^{\text{g}}}\right]_+.$$

Substituting the definition of $E_T$ completes the proof.

## F. Implementation details and hyperparameters

All experiments share the same prequential OPP runner and evaluation protocol; Table 2 summarizes shared settings, while Table 3 lists dataset-specific overrides (including auditing cadence and which COPF modules are enabled). Unless stated otherwise, *Base* disables auditing (audit off), whereas *Base+COPF* enables auditing and decision-layer control according to the schedule in Table 3.

*Table 2.* **Key hyperparameters (shared).** Unless otherwise noted in Table 3.

| Item | Value |
|------|-------|
| Protocol phases | Pre/Deploy/Post, each 20k rounds (total 60k). |
| Candidates | 1 positive + $N$=200 negatives (w/o replacement); if the positive is sampled as a negative it is removed, so $|C_t| \leq 201$. |
| Exposure policy | TOPK-STOCHASTIC, slate size $K$=10. |
| Exploration schedule | Pre: $\epsilon$=0.20, temp = 1.0; Deploy/Post: $\epsilon$=0.02, temp = 0.7. |
| Logging / windows | log every 1,000 rounds; OI window = 50,000. |
| Metrics | Mean±std over 3 seeds; worst-case-in-phase in parentheses. |

*Table 3.* **Dataset-specific overrides.** Shared settings are in Table 2.

| Dataset | Groups / outcome | Base | Base+COPF | Backbone notes |
|---|---|---|---|---|
| Synth-bip. | group_on = src; outcome = bandit | audit off | audit=1000; pre-cal; PD+cov in Deploy/Post | GraphMixer: tokens=neighbors. |
| TGBL-Wiki | group_on = dst; attr = node_mod(2) | audit off | audit=200; cf_upd=200; pre-cal; PD+cov in Deploy/Post | GraphMixer: neighbors=10, tokens=20 (stable). |
| TGBL-Review | group_on = dst; attr = node_mod(2) | audit off | audit=200; cf_upd=200; pre-cal; PD+cov in Deploy/Post | Same as Wiki. |

# G. Phase-wise summary tables

*Table 4.* **Wiki results (utility + counterfactual fairness).** Phase-wise metrics (mean $\pm$ std over seeds; worst-case in phase in parentheses).

| Backbone | Method | Phase | Utility | | | | Fairness | |
|---|---|---|---|---|---|---|---|---|
| | | | NDCG@10 | MRR | Hits@10 | DeployHit@TopK | $g_{\mathrm{gap}}^{\mathrm{TE}}$ | $g_{\max}^{\mathrm{Cal}}$ |
| EdgeBank | Base | PRE | 0.4085±0.0106 (0.3222) | 0.4127±0.0106 (0.3275) | 0.4248±0.0103 (0.3397) | 0.0755±0.0017 (0.0710) | 0.0109±0.0027 (0.0590) | 0.0241±0.0001 (0.0297) |
| | | DEPLOY | 0.5304±0.0136 (0.4911) | 0.5334±0.0134 (0.4966) | 0.5438±0.0138 (0.4990) | 0.1210±0.0061 (0.1133) | 0.0109±0.0037 (0.0423) | 0.0244±0.0002 (0.0296) |
| | | POST | 0.5886±0.0085 (0.5362) | 0.5909±0.0084 (0.5383) | 0.6014±0.0085 (0.5513) | 0.1324±0.0016 (0.1245) | 0.0073±0.0035 (0.0226) | 0.0248±0.0002 (0.0296) |
| | Base+COPF | PRE | 0.4085±0.0106 (0.3222) | 0.4127±0.0106 (0.3275) | 0.4248±0.0103 (0.3397) | 0.0754±0.0016 (0.0708) | 0.0109±0.0027 (0.0590) | 0.0241±0.0001 (0.0297) |
| | | DEPLOY | 0.5304±0.0136 (0.4911) | 0.5334±0.0134 (0.4966) | 0.5438±0.0138 (0.4990) | 0.1207±0.0061 (0.1130) | **0.0108±0.0040** (0.0431) | 0.0244±0.0002 (0.0296) |
| | | POST | 0.5886±0.0085 (0.5362) | 0.5909±0.0084 (0.5383) | 0.6014±0.0085 (0.5513) | 0.1321±0.0014 (0.1242) | 0.0074±0.0035 (0.0249) | 0.0248±0.0002 (0.0296) |
| TGN | Base | PRE | 0.3018±0.0125 (0.2854) | 0.2777±0.0116 (0.2545) | 0.4068±0.0162 (0.3439) | 0.0521±0.0010 (0.0509) | 0.0094±0.0049 (0.0528) | 0.3814±0.0005 (0.5054) |
| | | DEPLOY | 0.2335±0.0066 (0.2260) | 0.1935±0.0086 (0.1858) | 0.4037±0.0094 (0.3920) | 0.0506±0.0025 (0.0493) | 0.0135±0.0024 (0.0486) | 0.4390±0.0140 (0.5063) |
| | | POST | 0.2219±0.0049 (0.1868) | 0.1819±0.0018 (0.1525) | 0.3982±0.0155 (0.3473) | 0.0513±0.0010 (0.0502) | 0.0090±0.0029 (0.0226) | 0.4727±0.0139 (0.5090) |
| | Base+COPF | PRE | 0.3012±0.0019 (0.2822) | 0.2777±0.0028 (0.2480) | 0.4040±0.0052 (**0.3463**) | **0.0576±0.0025 (0.0520)** | **0.0082±0.0062 (0.0217)** | **0.3705±0.0138 (0.5000)** |
| | | DEPLOY | **0.2365±0.0059 (0.2295)** | 0.1918±0.0047 (0.1854) | **0.4221±0.0100 (0.4116)** | **0.0519±0.0020 (0.0510)** | **0.0115±0.0045** (0.0493) | 0.4413±0.0265 (0.5728) |
| | | POST | **0.2296±0.0082 (0.2115)** | **0.1872±0.0094 (0.1740)** | **0.4121±0.0060 (0.3747)** | **0.0537±0.0025 (0.0523)** | **0.0067±0.0025** (0.0242) | 0.4892±0.0121 (0.5725) |
| | Adv | PRE | 0.3018±0.0125 (0.2817) | 0.2777±0.0116 (0.2545) | 0.4068±0.0162 (0.3346) | 0.0521±0.0010 (0.0487) | 0.0094±0.0049 (0.0900) | 0.3814±0.0005 (0.5067) |
| | | DEPLOY | 0.2335±0.0066 (0.2230) | 0.1935±0.0086 (0.1842) | 0.4037±0.0094 (0.3819) | 0.0506±0.0025 (0.0459) | 0.0135±0.0024 (0.0939) | 0.4390±0.0140 (0.5120) |
| | | POST | 0.2219±0.0049 (0.1859) | 0.1819±0.0018 (0.1509) | 0.3982±0.0155 (0.3351) | 0.0513±0.0010 (0.0489) | 0.0090±0.0029 (0.0765) | 0.4727±0.0139 (0.5127) |
| | Adv+COPF | PRE | 0.3012±0.0019 (0.2764) | 0.2777±0.0028 (0.2480) | 0.4040±0.0052 (**0.3370**) | **0.0576±0.0025 (0.0511)** | **0.0082±0.0062 (0.0342)** | **0.3705±0.0138 (0.5036)** |
| | | DEPLOY | **0.2365±0.0059 (0.2257)** | 0.1918±0.0047 (0.1830) | **0.4221±0.0100 (0.3973)** | 0.0519±0.0020 (0.0469) | **0.0115±0.0045 (0.0325)** | 0.4413±0.0265 (0.5589) |
| | | POST | **0.2296±0.0082 (0.2017)** | **0.1872±0.0094 (0.1653)** | **0.4121±0.0060 (0.3505)** | **0.0537±0.0025 (0.0494)** | **0.0067±0.0025 (0.0193)** | 0.4892±0.0121 (0.5808) |
| | Reweight | PRE | 0.3020±0.0127 (0.2821) | 0.2776±0.0118 (0.2535) | 0.4076±0.0158 (0.3348) | 0.0521±0.0010 (0.0486) | 0.0091±0.0039 (0.0758) | 0.3814±0.0005 (0.5066) |
| | | DEPLOY | 0.2334±0.0067 (0.2231) | 0.1933±0.0087 (0.1840) | 0.4039±0.0095 (0.3821) | 0.0507±0.0025 (0.0457) | 0.0124±0.0016 (0.0750) | 0.4390±0.0140 (0.5120) |
| | | POST | 0.2219±0.0049 (0.1852) | 0.1819±0.0018 (0.1503) | 0.3982±0.0156 (0.3346) | 0.0513±0.0010 (0.0489) | 0.0084±0.0022 (0.0490) | 0.4727±0.0139 (0.5127) |
| | Reweight+COPF | PRE | 0.2988±0.0027 (0.2737) | 0.2749±0.0038 (0.2444) | 0.4028±0.0096 (**0.3370**) | **0.0575±0.0026 (0.0512)** | **0.0077±0.0060 (0.0317)** | **0.3706±0.0138 (0.5036)** |
| | | DEPLOY | **0.2344±0.0058 (0.2242)** | 0.1896±0.0070 (0.1809) | **0.4205±0.0095 (0.3975)** | 0.0519±0.0020 (0.0469) | **0.0109±0.0042 (0.0307)** | 0.4418±0.0266 (0.5589) |
| | | POST | **0.2268±0.0044 (0.2007)** | **0.1843±0.0051 (0.1647)** | **0.4089±0.0103 (0.3504)** | **0.0537±0.0025 (0.0494)** | **0.0064±0.0024 (0.0176)** | 0.4905±0.0123 (0.5808) |
| | Penalty | PRE | 0.3018±0.0125 (0.2817) | 0.2777±0.0116 (0.2545) | 0.4068±0.0162 (0.3346) | 0.0521±0.0010 (0.0487) | 0.0094±0.0049 (0.0900) | 0.3814±0.0005 (0.5067) |
| | | DEPLOY | 0.2335±0.0066 (0.2230) | 0.1935±0.0086 (0.1842) | 0.4037±0.0094 (0.3819) | 0.0506±0.0025 (0.0459) | 0.0135±0.0024 (0.0939) | 0.4390±0.0140 (0.5120) |
| | | POST | 0.2219±0.0049 (0.1859) | 0.1819±0.0018 (0.1509) | 0.3982±0.0155 (0.3351) | 0.0513±0.0010 (0.0489) | 0.0090±0.0029 (0.0765) | 0.4727±0.0139 (0.5127) |
| | Penalty+COPF | PRE | 0.3012±0.0019 (0.2764) | 0.2777±0.0028 (0.2480) | 0.4040±0.0052 (**0.3370**) | **0.0576±0.0025 (0.0511)** | **0.0082±0.0062 (0.0342)** | **0.3705±0.0138 (0.5036)** |
| | | DEPLOY | **0.2365±0.0059 (0.2257)** | 0.1918±0.0047 (0.1830) | **0.4221±0.0100 (0.3973)** | 0.0519±0.0020 (0.0469) | **0.0115±0.0045 (0.0325)** | 0.4413±0.0265 (0.5589) |
| | | POST | **0.2296±0.0082 (0.2017)** | **0.1872±0.0094 (0.1653)** | **0.4121±0.0060 (0.3505)** | **0.0537±0.0025 (0.0494)** | **0.0067±0.0025 (0.0193)** | 0.4892±0.0121 (0.5808) |
| GraphMixer | Base | PRE | 0.4818±0.0086 (0.4168) | 0.4036±0.0072 (0.3370) | 0.7537±0.0130 (0.7109) | 0.0779±0.0056 (0.0745) | 0.0097±0.0007 (0.0295) | 0.7428±0.0226 (0.9800) |
| | | DEPLOY | 0.3327±0.0490 (0.3244) | 0.2690±0.0479 (0.2583) | 0.5987±0.0412 (0.5854) | 0.0835±0.0023 (0.0783) | 0.0099±0.0016 (0.0548) | 0.7723±0.0270 (0.9759) |
| | | POST | 0.3353±0.0645 (0.2800) | 0.2824±0.0618 (0.2317) | 0.5663±0.0580 (0.5050) | 0.0853±0.0053 (0.0821) | 0.0086±0.0028 (0.0204) | 0.7651±0.0215 (0.9676) |
| | Base+COPF | PRE | **0.4843±0.0100 (0.4156)** | 0.4036±0.0097 (0.3342) | **0.7624±0.0078 (0.7141)** | **0.0825±0.0020 (0.0757)** | **0.0081±0.0003 (0.0284)** | 0.7644±0.0164 (0.9800) |
| | | DEPLOY | 0.3060±0.0151 (0.2958) | 0.2428±0.0156 (0.2369) | 0.5750±0.0095 (0.5521) | **0.0855±0.0003 (0.0821)** | 0.0104±0.0005 (**0.0424**) | 0.7752±0.0084 (0.9800) |
| | | POST | 0.2936±0.0378 (0.2401) | 0.2434±0.0349 (0.1973) | 0.5260±0.0411 (0.4545) | 0.0804±0.0036 (0.0775) | **0.0071±0.0019** (0.0231) | **0.7255±0.0214** (0.9706) |

*Table 5.* **Synthetic results (utility + counterfactual fairness).** Phase-wise metrics (mean ± std over seeds; worst-case in phase in parentheses).

| Backbone | Method | Phase | Utility | | | | Fairness | |
|---|---|---|---|---|---|---|---|---|
| | | | NDCG@10 | MRR | Hits@10 | DeployHit@TopK | $g_{\mathrm{gap}}^{\mathrm{TE}}$ | $g_{\mathrm{max}}^{\mathrm{Cal}}$ |
| EdgeBank | Base | PRE | 0.0445±0.0011 (0.0227) | 0.0507±0.0011 (0.0280) | 0.0713±0.0011 (0.0532) | 0.0488±0.0012 (0.0423) | 0.0073±0.0020 (0.0319) | 0.0210±0.0003 (0.0234) |
| | | DEPLOY | 0.1641±0.0047 (0.1317) | 0.1692±0.0045 (0.1369) | 0.1887±0.0050 (0.1577) | 0.0681±0.0042 (0.0581) | 0.0062±0.0008 (0.0219) | 0.0221±0.0002 (0.0241) |
| | | POST | 0.2777±0.0035 (0.2479) | 0.2800±0.0036 (0.2506) | 0.3049±0.0033 (0.2759) | 0.0861±0.0030 (0.0784) | 0.0088±0.0020 (0.0312) | 0.0231±0.0007 (0.0264) |
| | Base+COPF | PRE | 0.0445±0.0011 (0.0227) | 0.0507±0.0011 (0.0280) | 0.0713±0.0011 (0.0532) | 0.0488±0.0012 (0.0423) | **0.0068±0.0014** (0.0319) | 0.0210±0.0003 (0.0234) |
| | | DEPLOY | 0.1641±0.0048 (0.1317) | 0.1692±0.0046 (0.1369) | 0.1887±0.0051 (0.1577) | 0.0680±0.0040 (0.0579) | **0.0060±0.0007** (0.0221) | 0.0221±0.0002 (0.0241) |
| | | POST | 0.2777±0.0036 (0.2479) | 0.2800±0.0036 (0.2506) | 0.3049±0.0033 (0.2759) | 0.0858±0.0030 (0.0783) | **0.0085±0.0017** (0.0312) | 0.0231±0.0007 (0.0264) |
| TGN | Base | PRE | 0.0495±0.0024 (0.0225) | 0.0496±0.0022 (0.0279) | 0.0980±0.0034 (0.0533) | 0.0474±0.0013 (0.0418) | 0.0081±0.0015 (0.0299) | 0.3202±0.0109 (0.5004) |
| | | DEPLOY | 0.1414±0.0024 (0.1280) | 0.1211±0.0018 (0.1102) | 0.2516±0.0062 (0.2304) | 0.0478±0.0015 (0.0426) | 0.0081±0.0004 (0.0296) | 0.5135±0.0117 (0.5840) |
| | | POST | 0.1851±0.0022 (0.1700) | 0.1432±0.0031 (0.1360) | 0.3570±0.0018 (0.3120) | 0.0540±0.0043 (0.0503) | 0.0069±0.0014 (0.0285) | 0.5522±0.0146 (0.6714) |
| | Base+COPF | PRE | 0.0472±0.0012 (0.0226) | 0.0476±0.0009 (**0.0280**) | 0.0944±0.0032 (0.0530) | **0.0484±0.0022** (0.0417) | **0.0072±0.0025** (0.0236) | 0.3410±0.0464 (**0.4999**) |
| | | DEPLOY | **0.1442±0.0078** (**0.1334**) | **0.1224±0.0074** (**0.1146**) | **0.2553±0.0092** (**0.2334**) | 0.0530±0.0011 (0.0519) | 0.0092±0.0021 (0.0317) | 0.5422±0.0143 (0.6380) |
| | | POST | **0.1870±0.0083** (**0.1721**) | **0.1448±0.0067** (0.1360) | **0.3590±0.0111** (**0.3215**) | 0.0509±0.0016 (0.0472) | 0.0085±0.0012 (**0.0270**) | 0.6105±0.0039 (0.6923) |
| | Adv | PRE | 0.0495±0.0024 (0.0225) | 0.0496±0.0022 (0.0279) | 0.0980±0.0034 (0.0533) | 0.0474±0.0013 (0.0418) | 0.0081±0.0015 (0.0299) | 0.3202±0.0109 (0.5004) |
| | | DEPLOY | 0.1414±0.0024 (0.1280) | 0.1211±0.0018 (0.1102) | 0.2516±0.0062 (0.2304) | 0.0478±0.0015 (0.0426) | 0.0081±0.0004 (0.0296) | 0.5135±0.0117 (0.5840) |
| | | POST | 0.1851±0.0022 (0.1700) | 0.1432±0.0031 (0.1360) | 0.3570±0.0018 (0.3120) | 0.0540±0.0043 (0.0503) | 0.0069±0.0014 (0.0285) | 0.5522±0.0146 (0.6714) |
| | Adv+COPF | PRE | 0.0472±0.0012 (**0.0226**) | 0.0476±0.0009 (**0.0280**) | 0.0944±0.0032 (0.0530) | **0.0484±0.0022** (0.0417) | **0.0072±0.0025** (0.0236) | 0.3410±0.0464 (**0.4999**) |
| | | DEPLOY | **0.1442±0.0078** (**0.1334**) | **0.1224±0.0074** (**0.1146**) | **0.2553±0.0092** (**0.2334**) | 0.0530±0.0011 (0.0519) | 0.0092±0.0021 (0.0317) | 0.5422±0.0143 (0.6380) |
| | | POST | **0.1870±0.0083** (**0.1721**) | **0.1448±0.0067** (0.1360) | **0.3590±0.0111** (**0.3215**) | 0.0509±0.0016 (0.0472) | 0.0085±0.0012 (**0.0270**) | 0.6105±0.0039 (0.6923) |
| | Reweight | PRE | 0.0496±0.0024 (0.0225) | 0.0496±0.0022 (0.0279) | 0.0980±0.0036 (0.0533) | 0.0474±0.0013 (0.0418) | 0.0077±0.0018 (0.0307) | 0.3202±0.0109 (0.5004) |
| | | DEPLOY | 0.1414±0.0024 (0.1280) | 0.1211±0.0018 (0.1102) | 0.2516±0.0062 (0.2304) | 0.0478±0.0015 (0.0426) | 0.0081±0.0004 (0.0296) | 0.5135±0.0117 (0.5840) |
| | | POST | 0.1851±0.0022 (0.1700) | 0.1432±0.0031 (0.1360) | 0.3570±0.0018 (0.3120) | 0.0540±0.0043 (0.0503) | 0.0069±0.0014 (0.0285) | 0.5522±0.0146 (0.6714) |
| | Reweight+COPF | PRE | 0.0476±0.0016 (**0.0226**) | 0.0476±0.0006 (**0.0280**) | 0.0963±0.0053 (0.0530) | **0.0484±0.0014** (0.0417) | 0.0078±0.0025 (**0.0271**) | 0.3479±0.0200 (**0.4998**) |
| | | DEPLOY | 0.1418±0.0065 (**0.1310**) | 0.1208±0.0048 (**0.1133**) | 0.2514±0.0068 (**0.2341**) | 0.0530±0.0008 (0.0519) | 0.0087±0.0016 (0.0317) | 0.5426±0.0138 (0.6268) |
| | | POST | **0.1827±0.0102** (**0.1710**) | 0.1417±0.0043 (0.1344) | **0.3603±0.0112** (**0.3204**) | 0.0532±0.0018 (0.0490) | 0.0085±0.0008 (**0.0269**) | 0.6094±0.0059 (0.6923) |
| | Penalty | PRE | 0.0495±0.0024 (0.0225) | 0.0496±0.0022 (0.0279) | 0.0980±0.0034 (0.0533) | 0.0474±0.0013 (0.0418) | 0.0081±0.0015 (0.0299) | 0.3202±0.0109 (0.5004) |
| | | DEPLOY | 0.1414±0.0024 (0.1280) | 0.1211±0.0018 (0.1102) | 0.2516±0.0062 (0.2304) | 0.0478±0.0015 (0.0426) | 0.0081±0.0004 (0.0296) | 0.5135±0.0117 (0.5840) |
| | | POST | 0.1851±0.0022 (0.1700) | 0.1432±0.0031 (0.1360) | 0.3570±0.0018 (0.3120) | 0.0540±0.0043 (0.0503) | 0.0069±0.0014 (0.0285) | 0.5522±0.0146 (0.6714) |
| | Penalty+COPF | PRE | 0.0472±0.0012 (**0.0226**) | 0.0476±0.0009 (**0.0280**) | 0.0944±0.0032 (0.0530) | **0.0484±0.0022** (0.0417) | **0.0072±0.0025** (0.0236) | 0.3410±0.0464 (**0.4999**) |
| | | DEPLOY | **0.1442±0.0078** (**0.1334**) | **0.1224±0.0074** (**0.1146**) | **0.2553±0.0092** (**0.2334**) | 0.0530±0.0011 (0.0519) | 0.0092±0.0021 (0.0317) | 0.5422±0.0143 (0.6380) |
| | | POST | **0.1870±0.0083** (**0.1721**) | **0.1448±0.0067** (0.1360) | **0.3590±0.0111** (**0.3215**) | 0.0509±0.0016 (0.0472) | 0.0085±0.0012 (**0.0270**) | 0.6105±0.0039 (0.6923) |
| GraphMixer | Base | PRE | 0.1175±0.0018 (0.0555) | 0.1019±0.0004 (0.0569) | 0.2390±0.0044 (0.1013) | 0.0563±0.0014 (0.0417) | 0.0109±0.0021 (0.0367) | 0.7176±0.0183 (0.9796) |
| | | DEPLOY | 0.1417±0.0093 (0.1367) | 0.1250±0.0058 (0.1217) | 0.2952±0.0191 (0.2828) | 0.0787±0.0023 (0.0752) | 0.0103±0.0012 (0.0477) | 0.5816±0.0292 (0.9178) |
| | | POST | 0.1193±0.0059 (0.1087) | 0.1100±0.0043 (0.1021) | 0.2455±0.0101 (0.2268) | 0.0769±0.0066 (0.0706) | 0.0080±0.0005 (0.0329) | 0.5867±0.0214 (0.8321) |
| | Base+COPF | PRE | **0.1203±0.0018** (0.0539) | **0.1041±0.0023** (0.0555) | **0.2448±0.0010** (0.1015) | **0.0582±0.0017** (0.0467) | **0.0073±0.0015** (0.0434) | **0.6807±0.0038** (0.9091) |
| | | DEPLOY | **0.1996±0.0101** (**0.1926**) | **0.1604±0.0075** (**0.1564**) | **0.4154±0.0168** (**0.3999**) | **0.0800±0.0010** (0.0779) | **0.0076±0.0008** (0.0274) | **0.5337±0.0462** (0.7067) |
| | | POST | **0.1768±0.0006** (**0.1689**) | **0.1492±0.0002** (**0.1443**) | **0.3639±0.0016** (**0.3476**) | 0.0777±0.0037 (0.0725) | **0.0075±0.0007** (0.0274) | **0.5076±0.0089** (0.8037) |

*Table 6.* **Review results (utility + counterfactual fairness).** Phase-wise metrics (mean $\pm$ std over seeds; worst-case in phase in parentheses).

| Backbone | Method | Phase | Utility | | | | Fairness | |
|---|---|---|---|---|---|---|---|---|
| | | | NDCG@10 | MRR | Hits@10 | DeployHit@TopK | $g_{\mathrm{gap}}^{\mathrm{TE}}$ | $g_{\max}^{\mathrm{Cal}}$ |
| EdgeBank | Base | PRE | 0.0236±0.0002 (0.0215) | 0.0303±0.0002 (0.0279) | 0.0511±0.0006 (0.0465) | 0.0510±0.0014 (0.0445) | 0.0064±0.0048 (0.0760) | 0.0200±0.0000 (0.0200) |
| | | DEPLOY | 0.0210±0.0003 (0.0161) | 0.0281±0.0004 (0.0241) | 0.0460±0.0009 (0.0365) | 0.0488±0.0015 (0.0300) | 0.0059±0.0027 (0.0236) | 0.0200±0.0000 (0.0202) |
| | | POST | 0.0222±0.0006 (0.0175) | 0.0293±0.0006 (0.0250) | 0.0483±0.0010 (0.0420) | 0.0492±0.0018 (0.0410) | 0.0031±0.0002 (0.0135) | 0.0200±0.0000 (0.0202) |
| | Base+COPF | PRE | 0.0236±0.0002 (0.0215) | 0.0303±0.0002 (0.0279) | 0.0511±0.0006 (0.0465) | 0.0510±0.0014 (0.0445) | 0.0064±0.0048 (0.0760) | 0.0200±0.0000 (0.0200) |
| | | DEPLOY | 0.0210±0.0003 (0.0161) | 0.0281±0.0004 (0.0241) | 0.0460±0.0009 (0.0365) | 0.0488±0.0015 (0.0300) | 0.0059±0.0027 (0.0236) | 0.0200±0.0000 (0.0202) |
| | | POST | 0.0222±0.0006 (0.0175) | 0.0293±0.0006 (0.0250) | 0.0483±0.0010 (0.0420) | 0.0492±0.0018 (0.0410) | 0.0031±0.0002 (0.0135) | 0.0200±0.0000 (0.0202) |
| TGN | Base | PRE | 0.0352±0.0020 (0.0223) | 0.0416±0.0020 (0.0291) | 0.0626±0.0020 (0.0490) | 0.0512±0.0014 (0.0445) | 0.0067±0.0045 (0.0760) | 0.0427±0.0162 (0.5000) |
| | | DEPLOY | 0.0402±0.0015 (0.0324) | 0.0462±0.0014 (0.0377) | 0.0675±0.0020 (0.0605) | 0.0489±0.0015 (0.0310) | 0.0058±0.0025 (0.0236) | 0.0235±0.0018 (0.0734) |
| | | POST | 0.0389±0.0006 (0.0263) | 0.0444±0.0006 (0.0340) | 0.0683±0.0009 (0.0520) | 0.0495±0.0020 (0.0410) | 0.0030±0.0003 (0.0139) | 0.0420±0.0152 (0.5077) |
| | Base+COPF | PRE | 0.0350±0.0021 (0.0223) | 0.0414±0.0021 (0.0291) | 0.0624±0.0021 (0.0490) | 0.0512±0.0014 (0.0445) | 0.0068±0.0045 (0.0760) | **0.0426±0.0162 (0.4988)** |
| | | DEPLOY | 0.0399±0.0017 (0.0324) | 0.0458±0.0016 (0.0377) | 0.0672±0.0023 (0.0605) | 0.0489±0.0015 (0.0310) | **0.0056±0.0028 (0.0236)** | 0.0244±0.0026 (**0.0730**) |
| | | POST | **0.0392±0.0010 (0.0284)** | **0.0447±0.0011 (0.0362)** | **0.0687±0.0006 (0.0540)** | 0.0495±0.0018 (0.0410) | **0.0029±0.0003 (0.0140)** | 0.0409±0.0178 (**0.4998**) |
| | Adv | PRE | 0.0352±0.0020 (0.0223) | 0.0416±0.0020 (0.0291) | 0.0626±0.0020 (0.0490) | 0.0512±0.0014 (0.0445) | 0.0067±0.0045 (0.0760) | 0.0427±0.0162 (0.5000) |
| | | DEPLOY | 0.0402±0.0015 (0.0324) | 0.0462±0.0014 (0.0377) | 0.0675±0.0020 (0.0605) | 0.0489±0.0015 (0.0310) | 0.0058±0.0025 (0.0236) | 0.0235±0.0018 (0.0734) |
| | | POST | 0.0389±0.0006 (0.0263) | 0.0444±0.0006 (0.0340) | 0.0683±0.0009 (0.0520) | 0.0495±0.0020 (0.0410) | 0.0030±0.0003 (0.0139) | 0.0420±0.0152 (0.5077) |
| | Adv+COPF | PRE | 0.0350±0.0021 (0.0223) | 0.0414±0.0021 (0.0291) | 0.0624±0.0021 (0.0490) | 0.0512±0.0014 (0.0445) | 0.0068±0.0045 (0.0760) | **0.0426±0.0162 (0.4988)** |
| | | DEPLOY | 0.0399±0.0017 (0.0324) | 0.0458±0.0016 (**0.0377**) | 0.0672±0.0023 (0.0605) | 0.0489±0.0015 (0.0310) | **0.0056±0.0028 (0.0236)** | 0.0244±0.0026 (**0.0730**) |
| | | POST | **0.0392±0.0010 (0.0284)** | **0.0447±0.0011 (0.0362)** | **0.0687±0.0006 (0.0540)** | **0.0495±0.0018 (0.0410)** | **0.0029±0.0003 (0.0140)** | 0.0409±0.0178 (**0.4998**) |
| | Reweight | PRE | 0.0352±0.0021 (0.0223) | 0.0417±0.0021 (0.0291) | 0.0627±0.0021 (0.0490) | 0.0512±0.0014 (0.0445) | 0.0066±0.0047 (0.0760) | 0.0422±0.0162 (0.5000) |
| | | DEPLOY | 0.0403±0.0016 (0.0346) | 0.0462±0.0014 (0.0399) | 0.0679±0.0023 (0.0610) | 0.0489±0.0015 (0.0300) | 0.0060±0.0026 (0.0238) | 0.0253±0.0051 (0.0912) |
| | | POST | 0.0388±0.0008 (0.0280) | 0.0443±0.0007 (0.0356) | 0.0681±0.0014 (0.0530) | 0.0493±0.0020 (0.0410) | 0.0032±0.0007 (0.0137) | 0.0528±0.0223 (0.5105) |
| | Reweight+COPF | PRE | 0.0350±0.0022 (0.0223) | 0.0414±0.0022 (0.0291) | 0.0624±0.0022 (0.0490) | 0.0512±0.0014 (0.0445) | 0.0067±0.0047 (0.0760) | **0.0422±0.0162 (0.4988)** |
| | | DEPLOY | 0.0398±0.0017 (**0.0350**) | 0.0457±0.0015 (**0.0406**) | 0.0672±0.0023 (0.0595) | 0.0489±0.0015 (0.0300) | 0.0061±0.0025 (0.0238) | **0.0245±0.0026 (0.0731)** |
| | | POST | 0.0381±0.0009 (0.0262) | 0.0436±0.0010 (0.0328) | 0.0676±0.0006 (**0.0540**) | **0.0495±0.0019 (0.0410)** | 0.0033±0.0005 (0.0157) | **0.0318±0.0020 (0.1020)** |
| | Penalty | PRE | 0.0352±0.0020 (0.0223) | 0.0416±0.0020 (0.0291) | 0.0626±0.0020 (0.0490) | 0.0512±0.0014 (0.0445) | 0.0067±0.0045 (0.0760) | 0.0427±0.0162 (0.5000) |
| | | DEPLOY | 0.0402±0.0015 (0.0324) | 0.0462±0.0014 (0.0377) | 0.0675±0.0020 (0.0605) | 0.0489±0.0015 (0.0310) | 0.0058±0.0025 (0.0236) | 0.0235±0.0018 (0.0734) |
| | | POST | 0.0389±0.0006 (0.0263) | 0.0444±0.0006 (0.0340) | 0.0683±0.0009 (0.0520) | 0.0495±0.0020 (0.0410) | 0.0030±0.0003 (0.0139) | 0.0420±0.0152 (0.5077) |
| | Penalty+COPF | PRE | 0.0350±0.0021 (0.0223) | 0.0414±0.0021 (0.0291) | 0.0624±0.0021 (0.0490) | 0.0512±0.0014 (0.0445) | 0.0068±0.0045 (0.0760) | **0.0426±0.0162 (0.4988)** |
| | | DEPLOY | 0.0399±0.0017 (0.0324) | 0.0458±0.0016 (**0.0377**) | 0.0672±0.0023 (0.0605) | 0.0489±0.0015 (0.0310) | **0.0056±0.0028 (0.0236)** | 0.0244±0.0026 (**0.0730**) |
| | | POST | **0.0392±0.0010 (0.0284)** | **0.0447±0.0011 (0.0362)** | **0.0687±0.0006 (0.0540)** | **0.0495±0.0018 (0.0410)** | **0.0029±0.0003 (0.0140)** | 0.0409±0.0178 (**0.4998**) |
| GraphMixer | Base | PRE | 0.2350±0.0033 (0.1776) | 0.2183±0.0047 (0.1812) | 0.3089±0.0012 (0.1960) | 0.0596±0.0030 (0.0450) | 0.0107±0.0042 (0.0873) | 0.6209±0.0629 (0.9575) |
| | | DEPLOY | 0.2495±0.0048 (0.2395) | 0.2120±0.0073 (0.2009) | 0.3920±0.0038 (0.3706) | 0.0664±0.0001 (0.0600) | 0.0057±0.0015 (0.0221) | 0.7554±0.0656 (0.9800) |
| | | POST | 0.2301±0.0245 (0.1956) | 0.1963±0.0298 (0.1559) | 0.3681±0.0060 (0.3470) | 0.0592±0.0016 (0.0515) | 0.0066±0.0017 (0.0439) | 0.7242±0.0476 (0.9417) |
| | Base+COPF | PRE | 0.2322±0.0042 (**0.1781**) | 0.2141±0.0055 (0.1803) | **0.3103±0.0005 (0.2010)** | **0.0693±0.0017 (0.0510)** | **0.0049±0.0007 (0.0187)** | 0.6414±0.0360 (0.9800) |
| | | DEPLOY | **0.2511±0.0148 (0.2313)** | **0.2129±0.0188 (0.1915)** | **0.3951±0.0014 (0.3800)** | **0.0738±0.0011 (0.0698)** | 0.0074±0.0049 (0.1193) | **0.7479±0.0291** (0.9800) |
| | | POST | **0.2326±0.0238 (0.2017)** | **0.1985±0.0285 (0.1615)** | **0.3708±0.0056 (0.3571)** | **0.0620±0.0041 (0.0571)** | 0.0073±0.0038 (0.0545) | **0.6800±0.0105** (0.9762) |

## H. Spike Statistics and Certificate Slack

For each dataset–backbone pair, we report two phase-wise diagnostics. The spike statistic is the per-seed, per-phase maximum of the rolling maximum of $g_{\text{gap}}^{\text{TE}}$, using a window of $w = 5$ logged checkpoints. The certificate-slack statistic is the per-seed, per-phase median of

$$\frac{\widehat{B}_{\text{TE}}}{g_{\text{gap}}^{\text{TE}} + \varepsilon_{\text{num}}}, \qquad \varepsilon_{\text{num}} = 10^{-12},$$

where $\widehat{B}_{\text{TE}}$ is the logged Residual-OI upper bound on $g_{\text{gap}}^{\text{TE}}$. Values close to $1$ indicate tight certificates, whereas larger values indicate more conservative bounds.

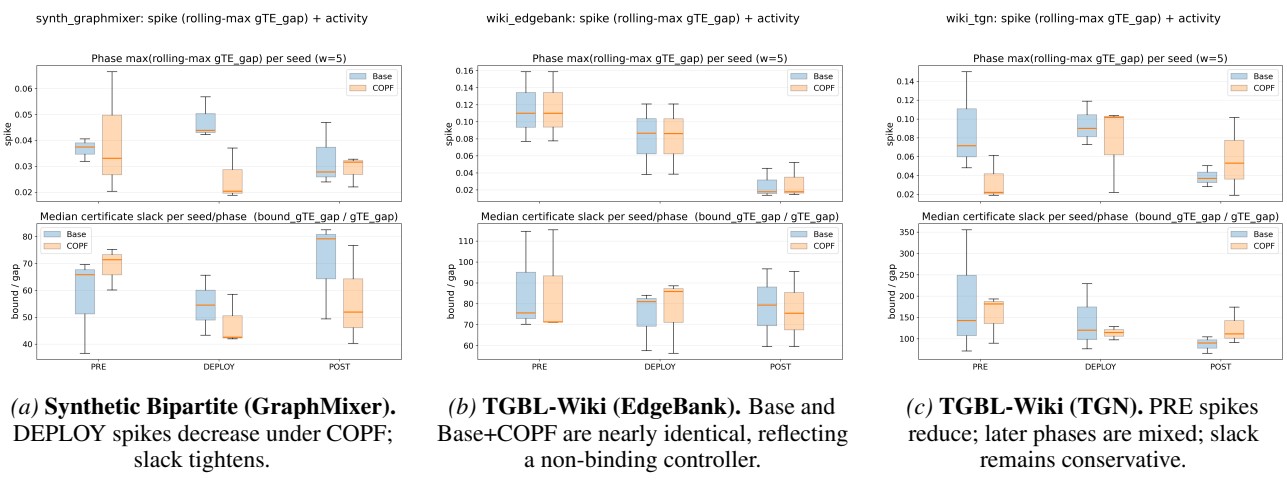

*(a)* **Synthetic Bipartite (GraphMixer).** DEPLOY spikes decrease under COPF; slack tightens.

*(b)* **TGBL-Wiki (EdgeBank).** Base and Base+COPF are nearly identical, reflecting a non-binding controller.

*(c)* **TGBL-Wiki (TGN).** PRE spikes reduce; later phases are mixed; slack remains conservative.

*Figure 4.* **Spike statistics and certificate slack (Base vs. COPF).** For each setting, the *top panel* reports phase-wise spikes via the per-seed maximum of the rolling maximum of $g_{\text{gap}}^{\text{TE}}$ over $w = 5$ logged checkpoints. The *bottom panel* reports median certificate slack, $\widehat{B}_{\text{TE}}/(g_{\text{gap}}^{\text{TE}} + \varepsilon_{\text{num}})$, per seed and phase, where $\widehat{B}_{\text{TE}}$ is the logged Residual-OI bound and $\varepsilon_{\text{num}} = 10^{-12}$ is used only for numerical stability.

## I. Propensity Logging Sensitivity

We conduct propensity-estimation stress tests on Synthetic+GraphMixer and TGBL-Wiki+TGN. The stress test varies the logged slate-entry propensity used by the counterfactual estimator while keeping the online prequential protocol, candidate stream, exposure policy, and evaluation windows fixed. We test whether moderate propensity perturbations induce abrupt failure or instead yield the continuous degradation predicted by the nuisance term $\varepsilon_e$ in Lemma 4.2 and Theorem 4.5.

Figure 5 reports two complementary sensitivity analyses. The IPS analysis isolates the effect of propensity perturbations under a fixed estimator family, while the GA-DR analysis evaluates the same pattern under the estimator used by COPF.

The fixed-IPS results show that moderate propensity perturbations do not qualitatively reverse the fairness or utility conclusions. Synthetic+GraphMixer remains highly stable across perturbations: Deploy and Post NDCG@10 are nearly invariant, while mean $g_{\text{gap}}^{\text{TE}}$ changes only slightly (Deploy: 0.001657–0.001876; Post: 0.001415–0.001656). TGBL-Wiki+TGN exhibits larger but still smooth variation rather than abrupt failure: Deploy NDCG@10 ranges from 0.213194 to 0.225652 and mean $g_{\text{gap}}^{\text{TE}}$ ranges from 0.001608 to 0.002166.

The GA-DR results show the same qualitative behavior across oracle/Monte Carlo and clipping variants. Synthetic+GraphMixer again varies smoothly, with Deploy mean $g_{\text{gap}}^{\text{TE}}$ ranging from 0.005486 to 0.009226, while TGBL-Wiki+TGN shows larger but still continuous degradation (Deploy mean $g_{\text{gap}}^{\text{TE}}$: 0.010825–0.015635). Overall, the results are consistent with continuous nuisance-error dependence rather than catastrophic sensitivity to moderate propensity misspecification.

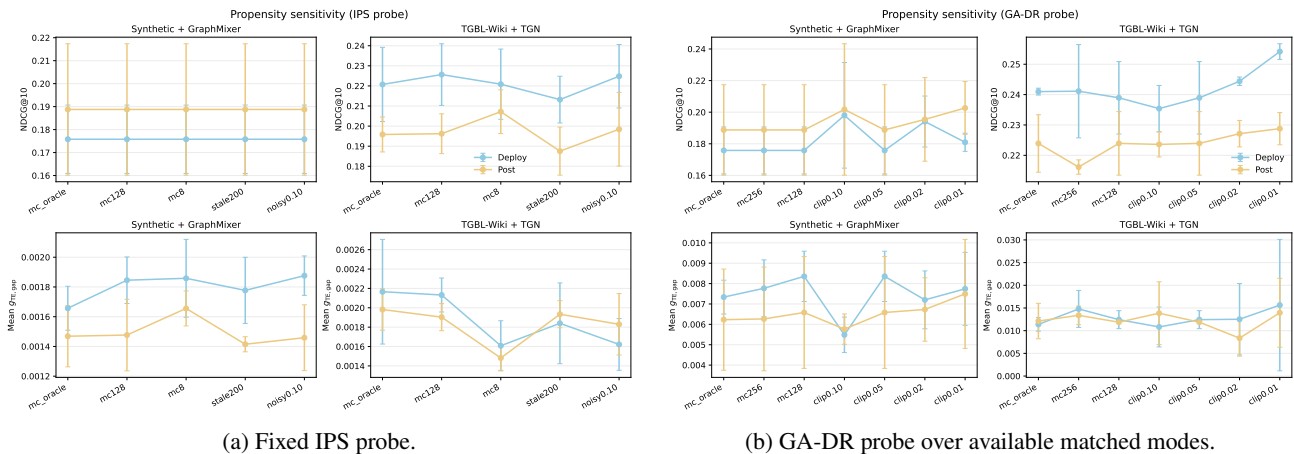

(a) Fixed IPS probe.      (b) GA-DR probe over available matched modes.

*Figure 5.* Sensitivity to propensity estimation under matched-run probes. Panel (a) fixes the estimator to IPS and varies oracle, lower-budget Monte Carlo, stale, and noisy logged propensities. Panel (b) reports the available GA-DR runs over oracle/Monte Carlo and propensity-clipping variants. Lines show Deploy and Post phase means; error bars show standard deviation across seeds when multiple seeds are available. The observed changes are continuous rather than abrupt, consistent with the $\varepsilon_e$-dependent nuisance terms in the GA-DR and Residual-OI transfer bounds.

## J. Target-Pair Feasibility of COPF

We include a feasibility diagnostic for the two active COPF constraints: the treatment-effect parity constraint and the minimum-effect guardrail. For a decision-layer configuration $\theta$, let

$$\widehat{\tau}_s(\theta) = \widehat{\mathbb{E}}_{\text{GA-DR}}\left[Y^{(1)} - Y^{(0)} \mid A = s\right]$$

be the estimated group treatment effect. COPF controls the treatment-effect gap

$$\widehat{g}^{\text{TE}}_{\text{gap}}(\theta) = \max_{s,s'} \left|\widehat{\tau}_s(\theta) - \widehat{\tau}_{s'}(\theta)\right|,$$

while also enforcing the minimum-effect guardrail

$$\widehat{g}^{\text{Min}}(\theta; \tau_{\min}) = \max_{s} \left[\tau_{\min} - \widehat{\tau}_s(\theta)\right]_+.$$

The former controls between-group disparity in exposure-induced benefit, whereas the latter prevents satisfying parity by suppressing benefit for all groups. Thus, the two constraints are complementary; infeasibility arises when the numerical target pair is outside the attainable frontier of the deployed decision layer.

Formally, let $\Theta$ denote the set of reachable decision-layer configurations. For a target pair $(\rho_{\text{TE}}, \tau_{\min})$, define the empirical feasible set

$$\widehat{\mathcal{C}}(\rho_{\text{TE}}, \tau_{\min}) = \left\{\theta \in \Theta : \widehat{g}^{\text{TE}}_{\text{gap}}(\theta) \le \rho_{\text{TE}}, \quad \min_{s} \widehat{\tau}_s(\theta) \ge \tau_{\min}\right\}.$$

The target pair is feasible if

$$\widehat{\mathcal{C}}(\rho_{\text{TE}}, \tau_{\min}) \neq \varnothing.$$

Equivalently, feasibility can be viewed through the attainable frontier

$$\widehat{\phi}(\rho_{\text{TE}}) = \max_{\theta \in \Theta: \widehat{g}^{\text{TE}}_{\text{gap}}(\theta) \le \rho_{\text{TE}}} \min_{s} \widehat{\tau}_s(\theta).$$

A pair is feasible when

$$\tau_{\min} \le \widehat{\phi}(\rho_{\text{TE}}),$$

up to the finite resolution of the sweep. This frontier view separates the conceptual compatibility of the constraints from the practical problem of choosing numerically attainable bounds.

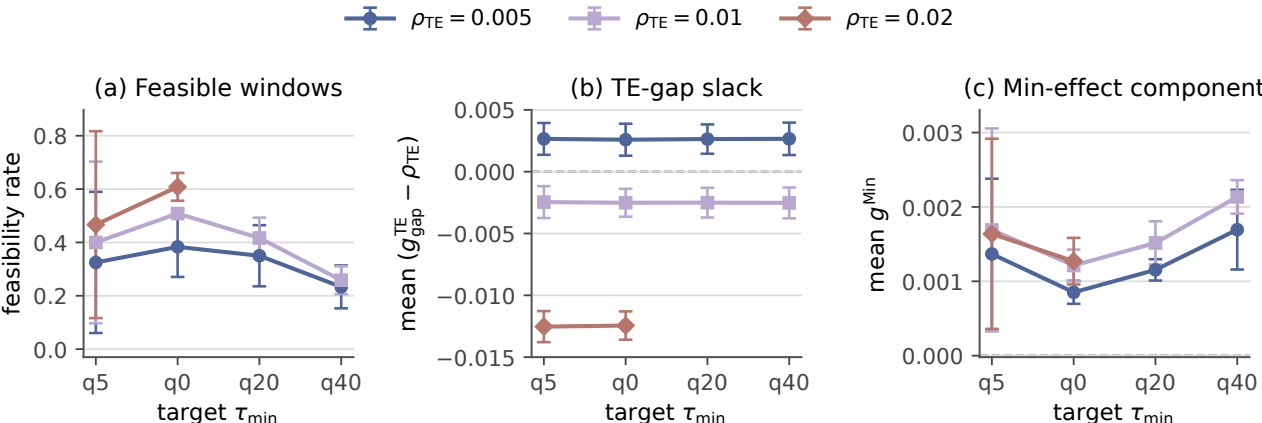

*Figure 6.* Target-pair feasibility diagnostic on Synthetic-Bipartite with GraphMixer at offset scale 1.0. Panel (a) reports the Deploy/Post feasible-window rate. Panel (b) reports the mean treatment-effect slack $g_{\text{gap}}^{\text{TE}} - \rho_{\text{TE}}$; the dashed zero line denotes exact satisfaction of the TE-gap target, and negative values satisfy the target. Panel (c) reports the mean minimum-effect violation $g^{\text{Min}}$, with zero denoting no violation. Error bars denote standard deviation across seeds. We plot completed non-imputed cells with three seeds for $\tau_{\min} \in \{q5, q0, q20, q40\}$ and $\rho_{\text{TE}} \in \{0.005, 0.01, 0.02\}$; unavailable cells, including the $\rho_{\text{TE}} = 0.02$ cells for $q20$ and $q40$ at this scale, are omitted rather than filled in.

For the empirical diagnostic, we evaluate feasibility on rolling Deploy/Post audit windows. Let $\mathcal{W}_{\text{DP}}$ be the set of such windows. We report

$$\widehat{R}_{\text{feas}}(\rho_{\text{TE}}, \tau_{\min}) = \frac{1}{|\mathcal{W}_{\text{DP}}|} \sum_{w \in \mathcal{W}_{\text{DP}}} \mathbf{1}\{\widehat{g}_{\text{gap},w}^{\text{TE}} \leq \rho_{\text{TE}}, \quad \widehat{g}_w^{\text{Min}} \leq 0\}.$$

This is an empirical target-setting diagnostic, not an exhaustive hyperparameter search. We run the diagnostic on Synthetic-Bipartite with GraphMixer, set $\tau_{\min}$ to selected pre-deployment treatment-effect quantiles, and sweep $\rho_{\text{TE}} \in \{0.005, 0.01, 0.02\}$. Figure 6 reports the completed scale-1.0 cells for the plotted target quantiles $\tau_{\min} \in \{q5, q0, q20, q40\}$, using three seeds per plotted cell. Cells not completed in the finite sweep are omitted rather than imputed.

Figure 6 shows the expected frontier behavior. Relaxing $\rho_{\text{TE}}$ moves the TE-gap slack below zero and increases the feasible-window rate. Conversely, increasing $\tau_{\min}$ from the baseline targets toward stricter nonnegative targets reduces feasibility, mainly because the minimum-effect guardrail becomes harder to satisfy. This supports the frontier interpretation: infeasibility is caused by choosing a numerical target pair outside the attainable empirical frontier, rather than by an inherent contradiction between treatment-effect parity and the minimum-effect guardrail.

The same frontier view gives a simple repair rule for target selection. If $\tau_{\min}$ is fixed, the smallest TE-gap tolerance supported by the decision layer is

$$\widehat{\rho}_{\text{TE}}^{\star}(\tau_{\min}) = \min_{\theta \in \Theta : \min_s \widehat{\tau}_s(\theta) \geq \tau_{\min}} \widehat{g}_{\text{gap}}^{\text{TE}}(\theta).$$

If $\rho_{\text{TE}}$ is fixed, the largest minimum-effect target supported by the decision layer is

$$\widehat{\tau}_{\min}^{\star}(\rho_{\text{TE}}) = \max_{\theta \in \Theta : \widehat{g}_{\text{gap}}^{\text{TE}}(\theta) \leq \rho_{\text{TE}}} \min_s \widehat{\tau}_s(\theta).$$

In the finite sweep, these quantities are approximated over the completed grid cells. Thus, when a proposed pair $(\rho_{\text{TE}}, \tau_{\min})$ is too strict, COPF can report the attainable frontier and relax one target while keeping the other fixed, rather than dropping either constraint.

