# OpenReview forum: "COPF: An Online Framework for Deployment-Stable Counterfactual Fairness in Evolving Graphs"
_ICML.cc/2026/Conference — ICML 2026 regular_

### Official Review · Reviewer_DWsF · 2026-03-10

**Soundness:** 3
**Presentation:** 3
**Significance:** 3
**Originality:** 3
**Overall Recommendation:** 4
**Confidence:** 3

**Summary:**

Online link recommendation systems on evolving graphs often exhibit performativity, because the links they expose affect future link formation and reshape the feedback data used for later evaluation, making fairness assessments based only on logged data vulnerable to distribution shift and fairness drift. To address this issue, this paper proposes COPF, a deployment stage framework for stable counterfactual fairness monitoring and control. COPF combines an online sequential protocol with explicit exploration and propensity logging, a graph aware doubly robust estimator for dynamic graph features, an auditing mechanism based on residual outcome indistinguishability, and a primal dual controller that adjusts exposure online. Theoretically, the paper proves a noisy transfer theorem showing that, under temporal mixing and bounded local interference, controlling estimated GA DR residuals can bound the true counterfactual group exposure gap. Experiments on synthetic bipartite graph streams and real benchmark datasets show that COPF substantially reduces extreme fluctuations in counterfactual group disparities in exposure while preserving ranking utility to a large extent.

**Compliance With Llm Reviewing Policy:**

Affirmed.

**Key Questions For Authors:**

1. When a large number of new nodes enter the system, how do the propensity logging mechanism in the online sequential protocol (OPP) and the graph-aware weight allocation mechanism adapt in order to avoid estimation failure?

2. The core theoretical transfer bound relies heavily on the assumptions of temporal mixing and bounded local interference. If these assumptions are violated in graph topologies with highly centralized nodes, what kind of degradation trend would COPF exhibit in terms of empirical robustness?

3. When transitioning from banditized feedback to real world settings that include partial organic feedback, that is, \(Y^{(0)} \neq 0\), does the pseudo outcome construction in the doubly robust estimator need to incorporate additional debiasing or baseline adjustment terms to ensure consistency?

**Strengths And Weaknesses:**

**Strengths**:

1. The paper makes an elegant connection between the counterfactual framework in causal inference and outcome indistinguishability in online learning. It not only proposes the GA DR estimation method, but also provides strong theoretical support for the framework through a transfer theorem that explicitly accounts for temporal dependence and local network interference in graph structures. In doing so, it helps fill a gap in the theoretical understanding of fairness in dynamic graphs.

2. As an independent decision layer wrapper, COPF can be integrated without modifying the core graph neural network model, and can be directly applied to systems such as TGN or GraphMixer. This nonparametric and lightweight intervention greatly reduces the practical friction of deploying advanced fairness algorithms in real world production environments.

3. The experimental design moves beyond the conventional practice of static dataset splitting and instead adopts a three stage online protocol consisting of Pre, Deploy, and Post phases. This dynamic process realistically captures the distribution shifts caused by recommendation policy updates, making the evaluation of deployment stability more concrete and much more aligned with real world system iteration scenarios.

**Weaknesses**:

1. Although the model is evaluated on TGB data streams, the lack of sensitive attributes in the benchmark datasets means that the experiments rely heavily on synthetic binary groupings based on node IDs, such as ID mod 2, as well as a strictly banditized feedback setting. This design may obscure the more complex forms of structural bias that exist in real social networks, as well as the dynamic complexity of organic graph evolution, where links form naturally rather than solely through platform driven exposure.

2. Although the system uses Monte Carlo sampling with truncation to estimate propensity scores and maintain numerical stability, it may be challenging to enforce bounded local interference and a positive lower bound on propensity scores, denoted by \(e_{\min}\), in real networks that commonly exhibit extreme long tail effects with a large number of low degree nodes. When the candidate set distribution is highly imbalanced, the variance of the GA--DR estimator may increase sharply.

---

> ### Author Rebuttal · Authors · 2026-03-31
>
> Thank you for the thoughtful and constructive feedback.
>
> **[W1]** Our empirical claim is not “full demographic fairness on organic social networks.” On TGB, ID-mod-2 is a placebo split and degree/activity is a structural sensitivity split; they test identifiability, auditing, and online control under OPP. The banditized outcome
> $$
> Y_t(u_t,v)=\mathbf 1\{v=v_t^\star\}D_t(v)
> $$
> is deliberate because it isolates policy-induced selection. The empirical signal: on the injected-bias synthetic stream with GraphMixer, COPF improves Deploy NDCG@10 from $0.1417$ to $0.1996$ while reducing $g_{gap}^{TE}$ from $0.0103$ to $0.0076$ and $g_{\max}^{Cal}$ from $0.5816$ to $0.5337$; on Wiki+TGN, the PRE worst-case $g^{TE}_{gap}$ drops from $0.0528$ to $0.0217$ (Tables 4-5). We will tighten the wording so the scope matches the evidence: deployment-stable, identifiable auditing/control of exposure-counterfactual fairness under performative selection.
>
> **[W2]** In our experiments, positivity is policy-enforced, not created by Monte Carlo truncation. Under Top-$K$ stochastic logging,
> $$
> e_t(v)=(1-\epsilon_t)e_t^{PL}(v)+\epsilon_t e_t^{uni}(v).
> $$
> Monte Carlo is used only to approximate/log $e_t^{PL}$. Because scores are clipped to $[10^{-4},1-10^{-4}]$ and $|C_t|\le 201$, the Plackett--Luce branch has full support, and the inclusion probability is lower bounded by its first-draw probability:
> $$
> e_t^{PL}(v)\ge\frac{\exp(10^{-4}/\tau_t)}{\exp(10^{-4}/\tau_t)+(|C_t|-1)\exp((1-10^{-4})/\tau_t)}>0.
> $$
>
> Thus the reported $e_{\min}$ comes from finite candidate budgets + clipped full-support stochastic logging + bounded temperature; Monte Carlo/clipping enters the theory only through the nuisance term $\varepsilon_e$ in Lemma 4.2. Without these design choices, uniform overlap need not hold and GA-DR variance can grow sharply; this is why the theorem is stated conditionally on policy-enforced overlap and bounded local dependence.
>
> **[Q1]** COPF is a decision-layer wrapper, so node/embedding initialization is backbone-specific; estimation stability is handled at the logging/certificate level. Appendix D.3 shows that concentration depends on the effective sample size
> $$
> W_{\mathrm{eff}}=\frac{(\sum_{t\in W_k}w_t)^2}{\sum_{t\in W_k}w_t^2}.
> $$
> Hence a burst of new nodes mainly reduces $W_{\mathrm{eff}}$ or slice mass $p_{\min}$, so the certificate widens rather than becoming falsely confident; coverage-driven exploration is exactly used to maintain the required $p_{\min}$.
>
> **[Q2]** The degradation is explicit in the bound:
> $$
> E_T=
> c_1\varepsilon_e\varepsilon_\mu
> +c_2(\varepsilon_e^2+\varepsilon_\mu^2)
> +c_3\sqrt{\frac{\tau_{\mathrm{mix}}\log|H|}{T}}
> +c_4\beta_T,
> \qquad
> \beta_W(\delta)=
> c\sqrt{\frac{\kappa\tau_{\mathrm{mix}}(\log|H|+\log(1/\delta))}{W_{\mathrm{eff}}}}
> +c'\varepsilon_{\mathrm{mix}}.
> $$
> Hub-dominated graphs enlarge the effective dependency degree $\kappa$ and typically worsen $(\tau_{\mathrm{mix}},\varepsilon_{\mathrm{mix}})$; thus certificates widen and fairness traces stabilize more slowly. If spillovers remain approximately local, degradation is gradual. If spillovers are genuinely long-range, the bounded-local-interference theorem is outside its validity regime. This locality/degree-control regime is standard in network-interference theory [1, 2, 5] and aligns with empirical spillover evidence in link recommendation [3, 4, 6].
>
> **[Q3]** No extra baseline term is needed. The DR pseudo-outcome already contains the arm-specific outcome regression $\mu_a(W_t)$, so when $Y^{(0)}\neq0$ one simply learns $\mu_0(W_t)=\mathbb E[Y_t^{(0)}\mid W_t]$. Moving from the banditized benchmark to partial organic feedback therefore changes the target regression, not the estimator form.
>
>
> **References**
>
> [1] Bayati et al. (NeurIPS 2024). Higher-order causal message passing for experimentation with complex interference.
>
> [2] Cortez et al. (NeurIPS 2022). Staggered rollout designs enable causal inference under interference without network knowledge.
>
> [3] Ferrara et al. (ACM Web Science Conference 2022). Link recommendations: Their impact on network structure and minorities.
>
> [4] Santos et al. (PNAS 2021). Link recommendation algorithms and dynamics of polarization in online
> social networks.
>
> [5] Leung (Econometrica, 2022). Causal inference under approximate neighborhood interference.
>
> [6] Su et al. (WWW 2016). The effect of recommendations on network structure.

---

> > ### Author Rebuttal · Reviewer_DWsF · 2026-04-03
> >
> > Thanks for your detailed reponse. I will stand by my original rating.

---

> > > ### Author Response · Authors · 2026-04-07
> > >
> > > Thank you for the encouraging feedback. We are glad that our response was helpful and that the clarifications addressed your concerns. We appreciate your careful reading and thoughtful comments.

---

### Official Review · Reviewer_cwPA · 2026-03-13

**Soundness:** 3
**Presentation:** 2
**Significance:** 2
**Originality:** 3
**Overall Recommendation:** 4
**Confidence:** 1

**Summary:**

This paper proposes COPF, a deployment-stable framework for fairness in evolving graphs. The authors model the decision as a treatment, and fairness across groups is measured by the treatment effect. The key idea behind the framework is to use propensity scores to overcome the performative problem. The authors further demonstrate the performance of the method through experiments.

**Compliance With Llm Reviewing Policy:**

Affirmed.

**Final Justification:**

The authors' rebuttal solved my concerns. I decide to maintain my score.

**Key Questions For Authors:**

1. How sensitive is COPF to propensity score estimation? I am interested in seeing experiments on different estimation methods.

2. It may strengthen the paper to clarify what happens if the assumptions do not hold, or in what settings they hold.

**Limitations:**

Yes

**Strengths And Weaknesses:**

### Strengths

1. Fairness learning in the performative setting is an important problem in reality setting.

2. The proposed method shows good perfermance in the experiments.

### Weaknesses

1. The framework might be sensitive to the estimation method of the propensity score.

2. (minor) The proposed framework uses causal inference techniques and thus some assumptions is needed (e.g. Assumption 3.1),  but these assumptions may not hold in practice.

---

> ### Author Rebuttal · Authors · 2026-03-30
>
> Thank you for the thoughtful and constructive feedback.
>
> **[W1]** We ran an additional matched-seed stress test (seed 42) over propensity modes {mc_oracle, mc128, mc32, mc8, stale200, noisy0.10} and estimator families {DM, IPS, DR, GA-DR}. While this is not a full multi-seed study, it is informative about whether COPF becomes brittle under moderate propensity perturbations. In this targeted check, we do not observe abrupt failure: on Synthetic+GraphMixer, holding the estimator fixed (IPS), Deploy/Post NDCG@10 remain $0.1797/0.1851$ across modes, while the mean $g_{\mathrm{TEgap}}$ error varies only from $0.00181$ to $0.00218$ in Deploy and from $0.00194$ to $0.00206$ in Post. On Wiki+TGN, the clearest effect is on Post utility/calibration under stale/noisy propensities (Post NDCG@10 changes from $0.2157$ under mc128 to $0.2043$ under stale200), while the mean $g_{\mathrm{TEgap}}$ error remains in a narrow $0.00135$--$0.00169$ band. So in this stress test, estimator choice appears to matter more than moderate propensity perturbation, and stale/noisy propensities mainly widen the certificate / modestly hurt utility rather than reversing the utility--fairness conclusion.
>
> This is also exactly what our theory predicts. In the proof of Lemma 4.2, the DR bias term satisfies
> $$
> \bigl| \mathbb E[h(W)\tilde\Gamma^{(a)}]-\mathbb E[h(W)Y^{(a)}]\bigr|
> \le \frac{\varepsilon_e\varepsilon_\mu}{e},
> $$
> and Theorem 4.5 / Corollary 4.6 yield
> $$
> \sup_{h\in H}\Bigl|\frac1T\sum_t h_t r_t\Bigr|
> \le
> \varepsilon_T
> +c_1\varepsilon_e\varepsilon_\mu
> +c_2(\varepsilon_e^2+\varepsilon_\mu^2)
> +c_3\sqrt{\frac{\tau_{\mathrm{mix}}\log|H|}{T}}
> +c_4\beta_T .
> $$
> So propensity misspecification enters continuously through nuisance error $(\varepsilon_e,\varepsilon_\mu)$ and overlap, rather than causing a qualitative reversal of the utility--fairness conclusion as long as exploration/clipping maintain support. We will add this ablation to the appendix.
>
> **[W2]** We agree the scope of the assumption should be stated more explicitly, but our intent is not to present overlap, local ignorability, temporal mixing, and bounded local interference as “free”; they are the structural conditions under which exposure-counterfactual quantities are identifiable from a single evolving, propensity-logged stream. In COPF these conditions are operationalized rather than left abstract: OPP-2 explicitly creates overlap by exploration/logging, $\hat e_t(v)\in[e_{\min},1-e_{\min}]$; Appendix C defines checkable finite-class mixing via empirical lag-$\ell$ covariances; and it defines computable local interference neighborhoods $I(u)$ and bounded dependency degree $\kappa$ from $G_{\le t}$ and the candidate rule.
>
> Each part of Assumption 3.1 matches closely related prior work. Overlap is standard in logged-bandit/OPE identification [7]. Local ignorability is the sequential analogue used in sequential OPE [3]. Bounded local interference/local dependence is the standard regime in network causal inference [2,1,4]. Finally, the substantive reason this matters in our setting is precisely that link recommendation changes local network evolution; this has been documented for “Who to Follow” style systems in [6].
>
> Just as importantly, if these assumptions weaken, our guarantee weakens in an explicit way rather than silently. Slower dependence enlarges the term $\sqrt{\tau_{\mathrm{mix}}\log|H|/T}$; misspecified propensity/outcome nuisances enter through $(\varepsilon_e,\varepsilon_\mu)$; weak support makes DR unstable, which is why exploration and clipping are built into OPP; and if important confounding remains outside $W_t$, then one needs confounded recommendation/OPE tools rather than plain DR [5,8]. We will make this “scope-of-validity / degradation-if-violated” point explicit in the main text, while keeping the central claim unchanged: COPF is a framework for identifiable and certifiable deployment-stable fairness in the regime where a single evolving log can support counterfactual auditing.
>
> **References**
>
> [1] Aronow & Samii (Ann. Appl. Statis. 2017). Estimating average causal effects under general interference.
>
> [2] Cortez et al. (NeurIPS 2022). Staggered rollout designs enable causal inference under interference without network knowledge.
>
> [3] Hu & Wager (Ann. Statist. 2023). Off-policy evaluation in partially observed Markov decision processes under sequential ignorability.
>
> [4] Leung (Econometrica 2022). Causal inference under approximate neighborhood interference.
>
> [5] Li et al. (NeurIPS 2023). Removing hidden confounding in recommendation: A unified multi-task learning approach.
>
> [6] Su et al. (WWW 2016). The effect of recommendations on network structure.
>
> [7] Swaminathan & Joachims (ICML 2015). Counterfactual risk minimization: Learning from logged bandit feedback.
>
> [8] Xu et al. (ICML 2023). An instrumental variable approach to confounded off-policy evaluation.

---

> > ### Author Rebuttal · Reviewer_cwPA · 2026-04-01
> >
> > Thanks for the rebuttal. I am satisfied with the answers.

---

> > > ### Author Response · Authors · 2026-04-07
> > >
> > > Thank you for the encouraging feedback. We are pleased that our response helped address your concerns. We also appreciate your constructive comments and suggestions.

---

### Official Review · Reviewer_zWQB · 2026-03-15

**Soundness:** 3
**Presentation:** 3
**Significance:** 3
**Originality:** 3
**Overall Recommendation:** 4
**Confidence:** 1

**Summary:**

This paper is proposed to address the fairness issue in online dynamic link recommendation systems. In such systems, there is an interation loop, where algorithmic decisions reshape the graph structure. This will create some challenges like performative prediction and deployment instability. In this setup, traditional fairness metrics could become misleading or drift after deployment. To resolve this, the authors propose the COPF framework, which includes two novel parts, the Graph-Aware Doubly Robust (GA-DR) estimators and Residual Outcome Indistinguishability (Residual-OI). This allows the algorithm to adaptively regulate node exposure via an online primal-dual controller. This paper also provides a theorem showing that auditing plug-in residuals online can effectively bound true counterfactual group gaps. They also empirically validate their algorithm using a full "Pre-Deploy-Post" lifecycle simulation on TGB datasets and synthetic bipartite streams. Their empirical results show their method maintains both good recommendation utility and fairness.

**Compliance With Llm Reviewing Policy:**

Affirmed.

**Final Justification:**

My concerns are addressed by the authors' reponses.

**Key Questions For Authors:**

Due to my limited knowledge in this area, I don't have very close related questions. However, I'm curious about the following questions

1. We have two constriants to ensure the benefit disparity and minimum effect guardrail are both bounded. The current theoretical analysis relies on feasibility assumption. Could the author elaborate on how to set these bounds and ensure two constraints are not conflicting for real world applications? If the bounds are not set properly, is it possible to find the nearest possible bound?

2. The fairness is ensured in expectation, which may not provide enough protection in the worst-case senarios. Is it possible to consider some more strict protections for the fairness?

**Limitations:**

yes

**Strengths And Weaknesses:**

Soundness
* Pros: The paper provides rigorous theoretical foundation for conterfactual analysis and online auditing based on residual outcome indistinguishability (OI). Compared with naive DR method, the porposed graph-aware doubly robust (GADR) estimators consider the unique properties in graphs. Their theoretical results ensure the noticable fairness gap ($h_t \cdot r_t$) is bounded in the long run.

* Cons: Their theoretical analysis based on some temporal mixing assumption. Such decayed effect is vital when they prove convergence of their residual OI. Though this is acceptable for the sake of temporal analysis, it could fundamentally limits the temporal depedency. In other words, if the real world case doesn't satisfy the mixing assumption, it's unclear whether such method could perform well. In addition, there may be some fundamental limitation in the bandit feedback. Some unmodeled factors (e.g. advertisement, searching) could also affect the edge generation and the GA-DR could face some systematic biases.

Presentation
* Pros: The paper is well organized with good motivation, clear problem setup, intuitive explaination on GA-DR and Residual-OI. The logic of the paper is very easy to follow and the theorem is clearly stated.

* Cons: The core algorithm (e.g. Algorithm 1) and the algorithm workflow is deferred to the appendix, which creates some barriers for non-experts. Also the notation could be further simplified with fewer subscripts and superscripts.


Significance
* Pros: Instead of considering a static recommendation problem with fairness regularier, this paper considers the fairness objective in an evolving graph. Such evolving graph better captures the real world user dynamics. This is a timely paper addressing the pain point in recommendation systems.

* Cons: As the algorithm operates on a sliding window and a few auditors are needed, it could be potentially expensive to implement the algorithm in large scale and highly dynamic systems. More detailed analysis on compute complexity would be preferred. In addition, the two constraints considered in the paper could conflict if the bound is not properly set. In real world problem, it's kind of challenging to choose such parameters and knowing the feasibility of such combination in advance.

Originality
* As I'm not an expert in this area, I don't have enough context to judge here. Based on this paper's description, I think the dynamic evolving graph is a novel setup and it's refreshing to see how fairness is ensured in such dynamic setup. However, the statistical tools seem to heavily rely on some well established frameworks (e.g. DR and OI). Though it's interesting to see such combination, I'm not sure about its fundamental contribution to the machine learning theory.

Small typos:
* Assumption 4.3 "confidence radii" should be "radis"?

> Disclaimer: This review is based on my expertise in online optimization and long-term fairness. While the paper is framed around fairness, its core technical components involve counterfactual analysis and evolving graphs, which are outside my main research areas. Therefore, my assessment may be limited, and I will consider other reviewers’ perspectives to further calibrate my evaluation.

---

> ### Author Rebuttal · Authors · 2026-03-30
>
> Thank you for the thoughtful and constructive feedback.
>
> **[soundness]** Our use of temporal mixing and bounded local interference is a dependence-control condition for deriving finite-sample counterfactual guarantees from a single logged evolving-graph stream, not a claim that every deployment literally has short memory. This role is explicit in Lemma 4.2 / Theorem 4.5. Also, the RHS of inequalities in Cor. 4.6 contains $E_T$. Hence if mixing is slower, the certificate weakens through $E_T$. This is standard in related sequential/network-causal theory: [2] give OPE rates that depend on mixing time, while [1] assume interference limited to low-degree neighbor interactions, which is stricter than our bounded-local-interference condition. Appendix C makes our assumptions operational via empirical lag-covariances and computable interference neighborhoods.
>
> The concern about ads/search is instead about local ignorability. The identifying claim is precisely that, if the main exposure/formation drivers are captured in $W_t$ and propensities are logged, GA-DR identifies the platform-mediated effect; otherwise hidden-confounder bias can remain, as in latent-confounded contextual bandits/OPE [3, 4, 5].
>
> **[significance]** Remark 4.7 states the asymptotic cost $O(|C_t|dL+|C_t|k+B)$ of an optimized incremental realization. The released runner is more conservative:
> $$
> T_t^{\mathrm{impl}}=T_{\mathrm{score}}(|C_t|)+T_{\mathrm{policy}}(|C_t|)+T_{\mathrm{DR}}(M_t)+\mathbf{1}\{t\bmod L_{\mathrm{audit}}=0\}T_{\mathrm{audit}}(W_t,|H|)+\mathbf{1}\{t\bmod L_{\log}=0\}T_{\mathrm{eval}}(W_t).
> $$
> The active-auditor routine updates only the top-$B$ violated auditors per window, although identifying them may still require scoring all auditors in $H$.
>
> On GraphMixer-synthetic with $N = 50,200,400$ negatives, mean ms/round are Pre 160.5/252.3/450.4, Deploy 1599.6/1786.6/1912.5, Post 1524.8/1887.3/1939.9. The key point is that amortized audit+OI+logging costs only 2.5–3.0 ms/round, while controller-side DR recomputation dominates Deploy/Post (about 1.44–1.61 s/round). Peak CPU RSS is 2.22/2.47/2.46 GB and peak GPU memory 186/394/654 MB. Thus the present bottleneck is controller-side DR estimation, not the sliding-window/active-auditor mechanism.
>
>
> **[Originality / presentation]** We do not claim DR or OI as new standalone primitives. The theoretical contribution is the noisy transfer from plug-in Residual-OI on GA-DR residuals to true counterfactual fairness gaps under temporal mixing and local interference, together with the formulation/identification/control pipeline for deployment-stable fairness in evolving graphs. For accessibility, we will place Algorithm 1 and the workflow figure in the main text and simplify notation.
>
> **[Q1]** Let $\tau_s(\pi)=\mathbb{E}[Y^{(1)}-Y^{(0)}\mid A=s;\pi]$. So $g_{\mathrm{TEgap}}(\pi)=\max_{s,s'}|\tau_s(\pi)-\tau_{s'}(\pi)|$, $g_{\mathrm{Min}}(\pi)=\max_s[\tau_{\min}-\tau_s(\pi)]_+$.
>
> Allowing tolerance $\eta\ge0$, write $L:=\tau_{\min}-\eta$, joint feasibility is equivalent to $\rho\ge\Psi(L)$, where $\Psi(L):=\inf_{\pi:\min_s\tau_s(\pi)\ge L} g_{\mathrm{TEgap}}(\pi)$.
>
> So infeasibility comes from asking for a pair $(\rho,L)$ outside the reachable frontier of the decision layer, not from an intrinsic contradiction between the two notions.
>
> A small deploy-phase sweep matches this frontier picture: stricter $\tau_{\min}$ lowers feasible-window rate (0.375 $\to$ 0.333 $\to$ 0.233 at $\rho_{\mathrm{TE}}=0.005$), while a looser $\rho_{\mathrm{TE}}$ partially restores feasibility.
>
> A conservative practical rule is to estimate attainable intervals $I_s=[\ell_s,u_s]$ and require $\max (L,\max_s(\ell_s-\rho))\le \min_s u_s$. If this fails, the nearest one-sided repairs are $\rho^{\mathrm{proj}}=$
> $\max \\\{\rho,(\max_s\ell_s-\min_su_s)_+\\\}$,
>
> $\tau_{\min}^{\mathrm{proj}}=\min\\\{\tau_{\min},\min_su_s\\\}$.
>
> **[Q2]** COPF does not only give a population-average statement. It audits $\sup_{h\in H}\big|\widehat{\mathbb{E}}[h\,r]\big|$ over an auditor family $H$, and Theorem 4.5 / Cor. 4.6 / Cor. D.5 give simultaneous high-probability control over all audited slices with mass at least $p_{min}$.
> Thus within the audited family the protection is worst-slice, not merely average-case; stronger per-dyad adversarial guarantees are a different target.
>
> **Reference**
>
> [1] Cortez et al. (NeurIPS 2022). Staggered rollout designs enable causal inference under interference without network knowledge.
>
> [2] Hu & Wager (Ann. Statist., 2023). Off-policy evaluation in partially observed Markov decision processes under sequential ignorability.
>
> [3] Li et al. (NeurIPS 2023). Removing hidden confounding in recommendation: A unified multi-task learning approach.
>
> [4] Sen et al. (AISTATS 2017). Contextual bandits with latent confounders: An NMF approach.
>
> [5] Xu et al. (ICML 2023). An instrumental variable approach to confounded off-policy evaluation.

---

> > ### Author Rebuttal · Reviewer_zWQB · 2026-04-01
> >
> > Thanks for your detailed reponse. My concerns have been adequately addressed

---

> > > ### Author Response · Authors · 2026-04-07
> > >
> > > Thank you for the encouraging feedback. We are glad that our response has adequately addressed your concerns. We also appreciate your thoughtful suggestions for improving the presentation.

---

### Decision · Program_Chairs · 2026-04-30

**Decision:**

Accept (regular)

**Comment:**

This paper studies fairness in online link prediction on evolving graphs, which is a decision-making problem. It proposes COPF, which includes a graph-aware doubly robust estimator, an auditor based on residual outcome indistinguishability, as well as a primal dual controller for exposure adjustment. Reviewers all appreciate the motivation for this paper, rigorous analysis, and good empirical performance. Reviewers' concerns are well addressed by the authors. Thus, I recommend acceptance.